# Impact of Topographic Wind Conditions on Dust Particle Size Distribution: Insights from a Regional Dust Reanalysis Dataset

Xinyue Huang[1], Wenyu Gao[2], Hosein Foroutan[1]

[1]Department of Civil and Environmental Engineering, Virginia Tech, Blacksburg, VA, 24060, USA
[2] Department of Mathematics and Statistics, University of North Carolina at Charlotte, Charlotte, NC, 28223, USA

*Correspondence to*: Hosein Foroutan (hosein@vt.edu)

**Abstract.** The size of windblown dust particles plays a critical role in determining their geochemical and climate impacts. This study investigates the relationship between topographic wind conditions (speed and direction relative to land slope) and the particle size distribution of dust emissions on a regional scale. We used the Multiscale Online Nonhydrostatic Atmosphere Chemistry (MONARCH) dust reanalysis dataset, which assimilates satellite data on coarse-mode dust optical depth ($DOD_{coarse}$). Validation against flight measurements from the 2011 Fennec campaign confirms the reanalysis's effectiveness in capturing coarse to super coarse dust. A 10-year dust reanalysis underwent selective screening to identify events with fresh emissions and the fraction of coarse dust concentrations was calculated as a surrogate for size distribution. The coarse fractions and associated meteorological and land characteristics obtained from various datasets were incorporated into multiple linear regression and machine learning models. Results indicate that dust particle size increases with wind speed, likely due to a higher fraction of fresh emissions and reduced deposition of coarse dust under stronger winds. A positive correlation between dust size and uphill slope suggests that enhanced vertical transport of dust by topography outweighs the impact of shifting emission microphysics over veering winds. Both positive correlations weaken in the afternoons and summer, likely due to the turbulence during haboob storms, which can suspend coarse dust from aged emissions, competing with the effect of uphill slopes. These findings on dust size dependency on topographic winds could improve representation of dust cycle and its impacts.

## 1 Introduction

Windblown dust particles emitted from arid and semi-arid areas are the largest terrestrial contributor to global aerosols (Brasseur and Jacob, 2017). Understanding the size of the airborne dust particles is crucial for assessing their impacts on global climate and biogeochemistry. Size plays an important role in determining the emission and deposition of dust particles and thus the redistribution of soil nutrients around the globe (Knippertz, 2017; Duce and Tindale, 1991; Wang et al., 2023). Additionally, particle size can affect the likelihood of microorganisms' attachment to dust aerosols (Polymenakou et al., 2008; Yamaguchi et al., 2012) and the distance these associated microorganisms can travel (Prospero et al., 2005; Kellogg and Griffin, 2006). Particle size, along with other factors such as mineralogy, chemical composition, and shape, controls the climate impacts of dust particles which can vary drastically from warming to cooling (Adebiyi and Kok, 2020; Kok et al., 2023; Mahowald et al., 2014). Realistic representation of dust particle size distribution (PSD) in the atmosphere requires an understanding of the dependencies of dust size at emission (Kok, 2011; Rosenberg et al., 2014).

Relationships between dust PSD and various environmental factors have been extensively studied. Soil moisture has been widely reported to increase the proportion of coarse dust in emissions by enhancing bonding forces among fine particles within soil aggregates (Dupont, 2022; González-Flórez et al., 2023; Shao et al., 2020), but one recent study also argues that the effect is non-

monotonic (Ma et al., 2023). Another surface characteristic of interest is soil texture. Emitted dust PSD during wind tunnel experiments was reported to be significantly influenced by the fully dispersed soil size distribution (Wang et al., 2021), and proportion of emitted submicron particles and $PM_{10}$ increased after tillage practices which broke down soil aggregates (Fernandes et al., 2023; Katra, 2020). The effect of atmospheric stability on dust size is inconsistent across studies. Some reported that an unstable atmospheric boundary is associated with richer submicron particles (Khalfallah et al., 2020; Shao et al., 2020), whereas others found no effect (Dupont, 2022; González-Flórez et al., 2023). Deposition impacts are universally acknowledged and were evaluated using characteristics of dust events or dust measurements, including fetch length (González-Flórez et al., 2023), dust age (Dupont et al., 2015; Ryder et al., 2013), and dust measurement height (Khalfallah et al., 2020; Shao et al., 2020). Nevertheless, the effects of multiple factors often intertwine during dust events, making the overall impact on dust size obscure.

Wind speed, or the resulting friction velocity ($u_*$) exerted on the ground surface, as the driving force of dust emissions, is one of the most essential parameters for dust emissions. The relationship between wind speed and PSD of dust emissions has been widely studied, yet consensus is lacking. Saltation-bombardment and aggregate disintegration are usually considered the primary mechanism for dust emissions (Kok et al., 2012). Parametrization of saltation-bombardment proposed that higher $u_*$ leads to higher energy in saltating particles and thus enhances the breaking down of soil aggregates and ejection of fine particles (Shao, 2001). This theory is supported by multiple wind tunnel experiments (Alfaro et al., 2022; Wang et al., 2021) and field measurements (Chkhetiani et al., 2021; Dupont, 2022; Khalfallah et al., 2020). The brittle fragmentation theory, on the other hand, postulates that PSD of vertical dust flux is independent of $u_*$, backed by compiled data from multiple wind tunnel and field measurements (Kok, 2011). Some also proposed that detachment of submicron particles from the surface of soil aggregates is more common when kinetic energy of impacting particles is low and the ejection of coarser particle from fragmentation becomes increasingly dominant with higher impaction intensity (Malinovskaya et al., 2021). These discrepancies are partially due to the interplay of other factors, such as inconsistencies in dust emission measurements (Khalfallah et al., 2020; Shao et al., 2020), soil moisture (Ishizuka et al., 2008; Shao et al., 2020; Webb et al., 2021), and whether steady state saltation is reached (Mahowald et al., 2014). Selection of the dust emission properties (e.g., dust flux or dust concentration; Shao et al., 2020) and the height of dust measurements (Khalfallah et al., 2020) can alter the dust PSD. The effects of soil moisture on shifting dust PSD at emission can get entangled with the potential effects of $u_*$. For instance, the fine fraction in dust emissions counterintuitively increasing with decreasing $u_*$ after light rain was caused by drying of the weakly crusted soils over time (Shao et al., 2020). With interferences of various factors, predicting the general dependency of PSD of dust emission on $u_*$ at regional scales over longer time becomes complex.

The role of topography in altering size of dust emission is of emerging interest. The orographic channelling of winds can affect the dust emission or transport (Caton Harrison et al., 2021; Rosenberg et al., 2014). Uphill winds can enhance the vertical transport of dust particles through flow separation, especially increase the proportion of coarse particles in the elevated dust based on computational simulations (Heisel et al., 2021). Moreover, the veering angle between wind vectors and the surface inclination can affect the emitted dust PSD. A study over a local field observed that compared to winds that blew more parallel to the ridges of the slopes (i.e., tangential winds), frontal uphill winds generated a higher fraction of fine particles (0.2-2 μm) because of more sputtering of fine particles on the windward slope due to resistance from the secondary aeolian structures, as well as less generation of coarse particles (2-5 μm) on the leeward slope with the recirculation zone (Malinovskaya et al., 2021). Other potential topographic impacts include the generation of erodible material by certain orographic winds (Washington et al., 2006) and the triggering of convective storms by mountains (Knippertz et al., 2007). However, their associations with dust PSD are understudied. Overall, it remains unclear whether the observed effects of wind over local topography on the PSD of dust emissions is detectable at regional scales.

Understanding the relationship between topographic wind conditions and PSD of dust emission on a regional scale is important

for simulating dust activities and impacts in atmospheric or climate models. Complementing the accumulating field data on PSD

of dust emissions (Shao et al., 2020; González-Flórez et al., 2023; Fernandes et al., 2023), this study aims to explore the impacts

of topographic wind conditions on dust PSD on a regional scale through data analysis. The regional scale means that the dust

emission of concern inevitably includes near-source transport and deposition. Here, we selected "fresh" dust emission events from

the Multiscale Online Nonhydrostatic AtmospheRe Chemistry model (MONARCH) dust reanalysis data (Di Tomaso et al., 2022)

and constructed models to investigate the correlations between PSD of surface dust concentrations in fresh emissions and wind

conditions over slopes, while taking into account other relevant meteorological and surface conditions. Our methodology,

sensitivity analysis, and evaluations are described in Section 2, and the discussion of our findings followed by the main takeaways

are presented in Sections 3 and 4.

## 2 Data and Methods

### 2.1 Datasets and variables

The study domain (12-38° N, 18W-36° E) encompasses the Sahara Desert, the largest dust source on Earth, which contributes to

around 60% of the global dust loading (Tanaka and Chiba, 2006). Various monitoring or reanalysis datasets are available for this

region, providing information on African dust sizes and the associated environmental conditions needed for this study.

The Multiscale Online Nonhydrostatic AtmoshpheRe Chemistry model (MONARCH) dust reanalysis (Di Tomaso et al., 2022)

dataset provides size-resolved dust information from 2007 to 2016 with 3-hour intervals. The dataset covers North Africa, the

Middle East, and Europe using a rotated-pole projection with a spatial resolution of 0.1°. The assimilation data of coarse-mode

dust optical depth ($DOD_{coarse}$) were derived from the Moderate Resolution Imaging Spectroradiometer (MODIS)-Aqua Deep Blue

level 2 aerosol products (Collection 6), including the aerosol optical depth (AOD), the Ångström exponent, and the single scattering

albedo at different wavelengths (Ginoux et al., 2012; Pu and Ginoux, 2016). MONARCH's first-guess dust size distribution follows

the brittle-fragmentation theory of Kok (2011) with perturbations across 12 ensemble members. By applying a local ensemble

transform Kalman filter with four-dimensional extension (4D-LETKF) at each 24-hour assimilation window, reanalysis increments

were added to the model ensemble simulations (first-guess) to match the $DOD_{coarse}$ observations. Specifically, the dust state vector

of the total coarse dust mixing ratio (distributed across five coarser bins from 1.2 to 20 µm) was updated, then the increments for

the finer three bins were determined proportionally to their total relative mass. Therefore, although MONARCH reanalysis does

not directly assimilate fine-mode DOD, corrections in the coarse bins propagated to the entire PSD through the assimilation state

vector and physical parameterizations, aligning the PSD more closely with dust-specific observations. Consequently, if the prior

PSD is biased—for instance by placing too much mass in the largest bin or not enough in a medium bin—that bias may persist to

some extent after assimilation. Despite the limitation, validation against observational data from the Aerosol Robotic Network

(AERONET) indicates that fine dust is still captured satisfactorily (Di Tomaso et al., 2021; Mytilinaios et al., 2023), supporting

the reliability of the dataset to investigate dust PSD. We hypothesize that the dust concentration reanalysis captures the potential

regional effects of topographic wind conditions on the dust PSD via assimilation of the satellite $DOD_{coarse}$ observations. The

adjustments in dust concentration PSD during data assimilation are showcased by  the uneven ratio of the first-guess dust

concentration to its reanalysis across eight size bins (see section 2.2 and Fig. S1). The MONARCH model was run in 40 hybrid

pressure-sigma model layers and the dust concentration in the lowest layer was saved as the surface dust concentration. The surface

dust concentration best represents near-source dust emissions and transport, and was chosen for this study. The concentration is available in eight size bins (i.e., 0.2–0.36, 0.36–0.6, 0.6–1.2, 1.2–2.0, 2.0–3.6, 3.6–6.0, 6.0–12.0, and 12.0–20.0 µm; Klose et al., 2021)). For easier comparison, these size-resolved concentrations were condensed into a single index called the "coarse fraction", defined as the sum of mass concentrations of the coarsest two bins (6-12 µm) divided by the total mass concentration of all eight bins (referred to as "cf2"). The delineation of fine and coarse particles is somewhat arbitrary and case-specific across studies. In general, a cutoff diameter from submicron to above 10 µm was used (Dupont, 2022; Fernandes et al., 2023; Panebianco et al., 2023) and the most common range is roughly 2-5 µm (Ishizuka et al., 2008; Ryder et al., 2019; Shao et al., 2020; Webb et al., 2021), which generally aligns with the lower boundary of 6 µm used in this study. Additionally, we tested alternative definitions of coarse fraction (namely "cf1" and "cf3", where the coarsest one or three bins were assigned as coarse dust) in the subsequent statistical analysis and results suggest that cf2 is a representative surrogate for dust particle size. More details on the comparison are included in Table S4 and Section 3.3.

Because wind conditions associated with dust concentrations are not available from the MONARCH dust reanalysis dataset, we sourced the information from the Modern-Era Retrospective analysis for Research and Applications (MERRA-2) data (Gelaro et al., 2017). MONARCH ensemble simulations applied meteorological inputs from two reanalysis datasets, i.e., MERRA-2 and ERA-Interim. Given that wind from both reanalyses are highly constrained by observations, and there is a substantial overlap in the assimilated data used by the two (Fujiwara et al., 2024; Rienecker et al, 2008; Dee et al., 2011; Gelaro et al., 2017), it is reasonable to use MERRA-2 wind vectors to inform the wind conditions of MONARCH dust reanalysis. The available wind components nearest to the surface and most relevant to dust emissions are at 2 meters above ground, provided as hourly average with a spatial resolution of 0.5° latitude × 0.625° longitude in the product M2I1NXASM. Wind speed and wind direction were subsequently calculated.

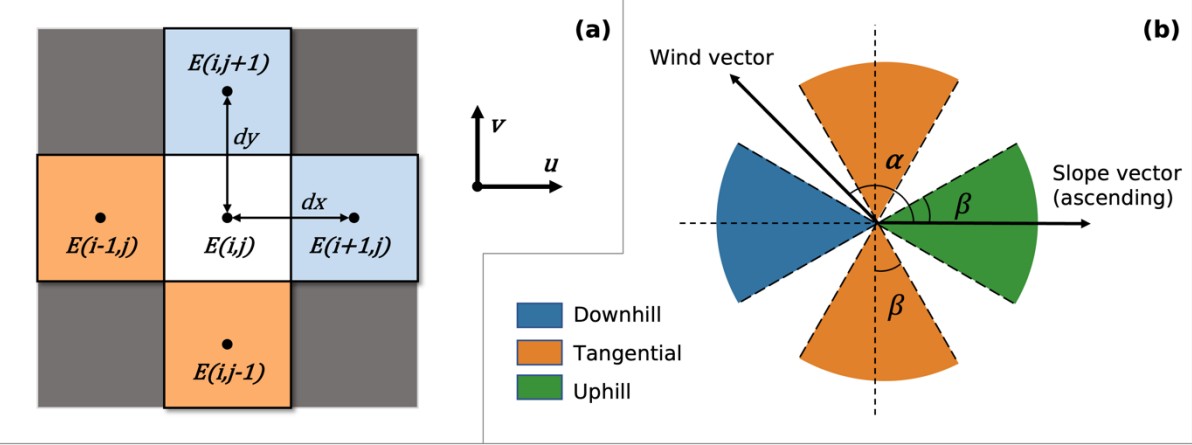

**Figure 1.** (a) Variables used in the calculation of the upwind slope. The elevations in the target grid, E(i,j) and the upwind grids (in orange if the wind components were positive; in light blue if the wind components were negative) were used to calculate the upwind slope components in x and y directions. (b) Methods for assigning the type of wind direction over topography. The angle between the upwind slope and the wind vector, α, and the predefined cut-off angle, β, together determined whether the wind from an event was typical for any of the three categories of relative wind direction, namely downhill, tangential, and uphill winds.

The wind direction types relative to topography (hereafter "relative wind direction type") were determined based on the wind vector (represented by $u(i,j)$ and $v(i,j)$) and the slope vector (represented by $S_x(i,j)$ and $S_y(i,j)$). Elevation data for calculating the slope were retrieved from the NASA Shuttle Radar Topography Mission Global 3 arc-second (SRTM GL3) dataset (Farr et al.,

2007). The SRTM elevation data are expected to be compatible with the MERRA-2 wind reanalysis because 1) both the SRTM and GTOPO30, which is used in the Goddard Earth Observing System (GEOS-5) model (MERRA-2's first-guess), have much finer resolution than MERRA-2's grids, making their spatially averages comparable, and 2) MERRA-2 wind reanalysis are highly constrained by assimilated observations, reducing its dependency on topographic input. Dust concentration is most relevant to emissions in upwind grids, therefore, slope vector components in each grid $(i,j)$ were determined as gradients of elevation between the target grid and the two neighboring grids in the upwind directions (Fig. 1(a) & Eq. (1)). To the best of the authors' knowledge, this study presents the first derivation of the upwind slope over North Africa.

The wind direction over topography was categorized into tangential, uphill, and downhill winds depending on the angle between the slope and wind vectors, $\alpha$ ($0 \leq \alpha < 360°$) as well as a predefined cut-off angle, $\beta$ ($0 < \beta \leq 45°$) (Fig. 1(b) & Eq. (2)). A smaller cut-off angle will result in a more selective process for assigning the relative wind direction types.

$$S_x(i,j) = \begin{cases} \frac{E(i,j)-E(i-1,j)}{dx}, & u(i,j) > 0 \\ \frac{E(i,j)-E(i+1,j)}{dx}, & u(i,j) < 0 \end{cases}, S_y(i,j) = \begin{cases} \frac{E(i,j)-E(i,j-1)}{dy}, & v(i,j) > 0 \\ \frac{E(i,j)-E(i,j+1)}{dy}, & v(i,j) < 0 \end{cases}, \tag{1}$$

where $(i,j)$ denotes the location of a grid cell, $E(i,j)$ represents the elevation, and $S_x(i,j)$ and $S_y(i,j)$ represent the slope components in $x$ and $y$ directions, respectively. The $u(i,j)$ and $v(i,j)$ are horizontal wind components.

$$\textbf{wind type} = \begin{cases} \text{uphill}, & \boldsymbol{0 < \alpha < \beta} \textbf{ or } \boldsymbol{(360° - \beta) < \alpha < 360°} \\ \text{tangential}, & \boldsymbol{(90° - \beta) < \alpha < (90° + \beta)} \textbf{ or } \boldsymbol{(270° - \beta) < \alpha < (270° + \beta)}, \\ \text{downhill}, & \boldsymbol{(180° - \beta) < \alpha < (180° + \beta)} \end{cases} \tag{2}$$

The land characteristics of soil texture and soil moisture that are expected to cast impacts on PSD of dust emissions were considered. Spatial distribution of soil texture was adopted from the map used in the Global Land Data Assimilation System version 2 (GLDAS2) Noah land surface model (Rodell et al., 2004), derived from the global soil dataset by Reynolds et al. (2000). The texture of the top layer of soil was categorized into the 16 classes developed by the Food and Agricultural Organization (FAO), varying in sand, silt, and clay fractions (Jahn et al., 2006). Soil moisture data were retrieved from the MERRA-2 product M2T1NXLND, providing average water content in the top 5-centimeter layer of soil hourly with a spatial resolution of 0.5° latitude × 0.625° longitude.

All datasets were co-registered onto a universal 0.1° Plate Carrée coordinate. MERRA-2 and MONARCH data were regridded using the Python xesmf package version 0.7.1 (Zhuang et al., 2023). The 2 m wind vectors and the soil moisture from the MERRA-2 reanalysis underwent upsampling using the "nearest source to destination" algorithm to match MONARCH's finer resolution. This algorithm did not bring in artificial variations so was the safest choice for regridding. Coarser spatial resolution of the MERRA-2 data meant some neighboring grids inevitably shared the same wind vector and soil moisture, diminishing the potential effects of wind conditions on dust PSD. The SRTM elevation data with a much higher original resolution were downsampled using the "average" method provided by the Python package of geowombat version 2.1.6 (Graesser, 2023). Sub-grid information on topography was lost, but handling topography information at the same scale as the wind data was reasonable because the terrain variations at finer resolution were deemed smooth when scaling up. Soil texture data at 0.25° lat-lon coordinates were also projected to the 0.1° coordinates. To match the instantaneous 3-hourly timesteps of the MONARCH reanalysis, we picked the average wind

components and soil moisture from the precedent hour, relevant to the initial dust emissions that could remain airborne during that
time period. Maps showing the average wind speed, the average upwind slope, the most common wind direction types, and the
average surface dust concentration at 0.1° resolution over the 10 years are presented in Fig. 2. While we acknowledge the inherent
resolution limitations of reanalysis datasets, the focus of this study is on the broader-scale modulation of dust emission by wind
conditions, and data assimilation combined with upsampling techniques ensure that our conclusion remain interpretable in this
context.

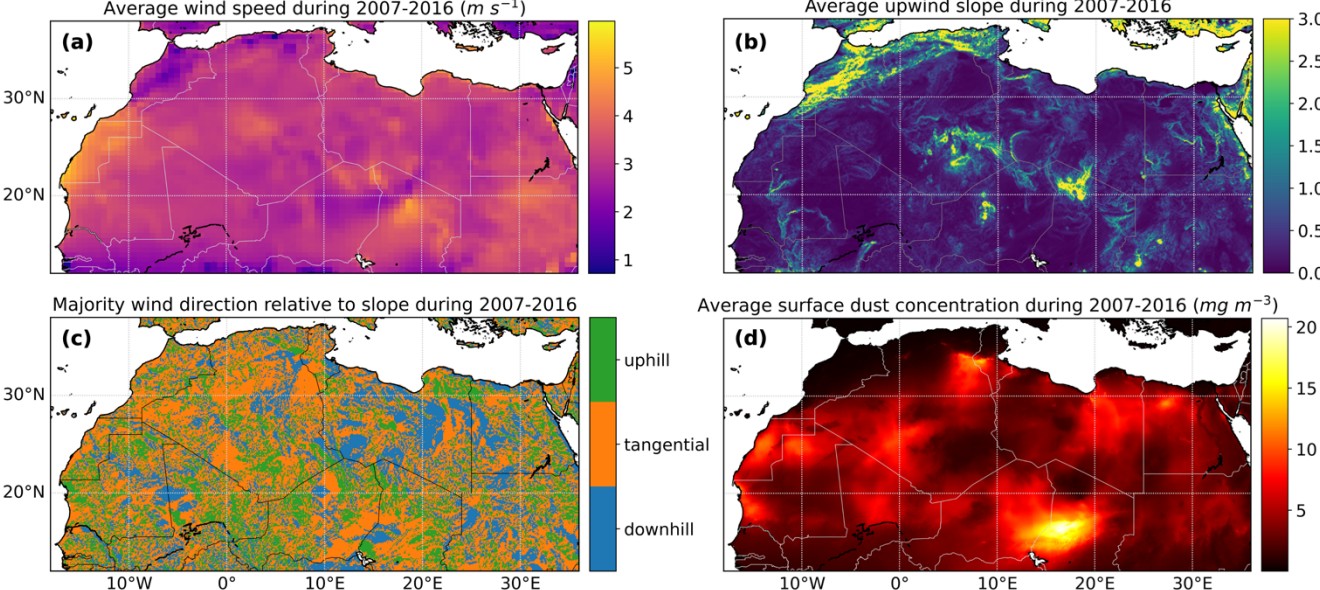

**Figure 2.** General spatial patterns of (a) 2m wind speed from the MERRA-2 reanalysis, (b) calculated upwind slope, (c) derived wind direction
type relative to slope using cut-off angle of 10°, and (d) MONARCH total dust concentration at ground surface from 2007 to 2016. The average
values are shown for wind speed, upwind slope, and dust surface concentration, and the most frequent types are shown for the relative wind
direction.

## 2.2 Validation against the Fennec measurements

The MONARCH dust reanalysis dataset was previously evaluated against observations from the AERONET retrievals (Di Tomaso
et al., 2021; Mytillinaios et al., 2023). Comparison between the 3-hourly MONARCH reanalysis of DOD$_{coarse}$ at 550 nm and the
coarse-mode AOD at 500 nm from AERONET retrievals over Sahara generate a Pearson correlation of 0.81 with a root mean
square error of 0.15 (Di Tomaso et al., 2022). Here, we present an additional case study to particularly evaluate the performance
of dust reanalysis on capturing fresh dust emissions. Observational data were obtained from the 2011 Fennec campaign, where
size-resolved dust emissions over western Africa were intensively sampled using wing-mounted instruments (Ryder et al., 2013).
Segments of three flights (b600–602), each lasted 10 minutes, over northern Mali on 17–18 June 2011, were identified to be
associated with fresh dust uplifts by low-level jets (Ryder et al., 2013; Ryder et al., 2015). Measured dust number concentration
during these flight segments was converted to volumetric concentration for easier comparison with the MONARCH reanalysis
data. The MONARCH reanalysis grids containing any portion of these flight trajectories were identified, and the associated dust
mass concentrations were retrieved. These concentrations were weighted averaged by flight duration in each grid cell to yield an
overall binned dust concentration. The MONARCH dust mass concentrations were also converted into volumetric concentrations
using the dust particle density of 2500 kg m$^{-3}$ for the finer four bins and 2,650 kg m$^{-3}$ for the coarser four bins (Klose et al., 2021).

As shown in Fig. 3, the trends of two dust PSDs generally agree well across all the eight size bins of the MONARCH dataset. Most notably, MONARCH reanalysis is effective at capturing the coarse to super coarse modes (defined as dust with diameter greater than 10 µm; Meng et al., 2022) represented by the last three bins (3.6–20 µm), outperforming several recent dust simulations that lack the data assimilation (Adebiyi and Kok, 2020; Meng et al., 2022). Additional investigations (see Fig. S1) revealed that the reanalysis dust concentration changes non-monotonically from its first-guess, leading to the conclusion that not only the total dust concentration but also its PSD has changed through assimilation. Overall, the predicted concentration for fresh emitted dust lies in an acceptable range, and the data assimilation process improved concentration across all bins as well as adjusted the dust size distribution, suggesting that the MONARCH reanalysis reasonably represents fresh dust emissions.

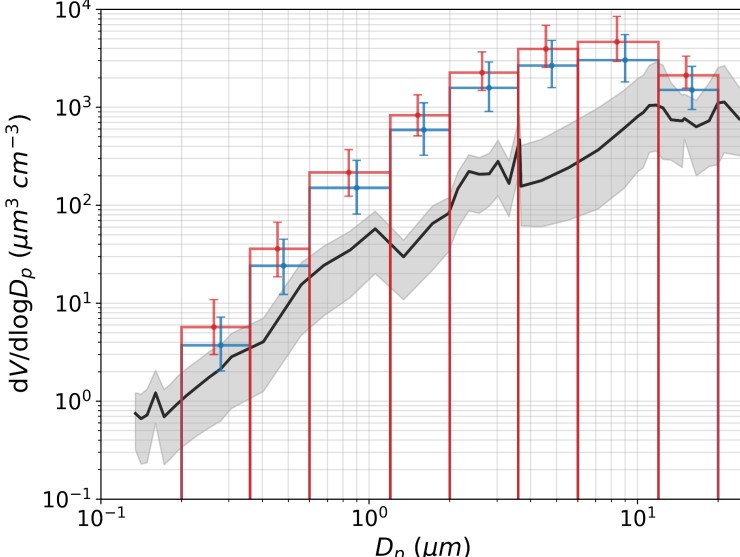

**Figure 3.** The black line shows the average volumetric concentration (µm$^3$ cm$^{-3}$) of dust sampled during three Fennec flights (6 flight segments) and the grey shaded area denotes the range of values. The blue bars show the volumetric concentration (µm$^3$ cm$^{-3}$) of dust in corresponding grids from the MONARCH reanalysis calculated from the weighted average mass concentration, with a particle density of 2,500 kg m$^{-3}$ for the finer four bins and 2,650 kg m$^{-3}$ for the coarser four bins. The error bars denote the range of values.

## 2.3 Event selection and sensitivity tests

Addressing the specific goals of this study requires the selection of the most relevant dust events from a decade of data over northern Africa. Two goals guided the screening process: 1) excluding the aged dust, and 2) focusing on wind conditions over distinctive terrain variations. Several screening criteria were accordingly developed that must be met simultaneously for dust events to be eligible for further analysis. Specifically, the selected dust events must occur within dust sources and over terrain with prominent slopes, be concurrent with high wind speeds and typical wind directions over slope, and have notable increases in dust surface concentrations. The procedures for screening these events are illustrated in Fig. 4 and described below in detail. This highly selective approach was made possible by the abundance of data from the MONARCH reanalysis.

To start with, dust events were confined to dust sources to exclude long-range transported and likely aged dust far from dust-source regions. The 10-year average of surface dust concentrations was calculated for the entire study domain, and pixels with values above the threshold percentile were designated as dust sources. For example, dust sources selected using the 80[th] percentile of the

10-year average dust concentration as the threshold are presented in Fig. 4(a). The map of dust sources features the Bodélé

Depression, Great Sand Sea, Tanezrouft, and the Atlantic Coastal Desert, consistent with the dust sources identified in other studies

(Formenti et al., 2011; Di Tomaso et al., 2022). Further screening steps were performed independently within these dust sources.

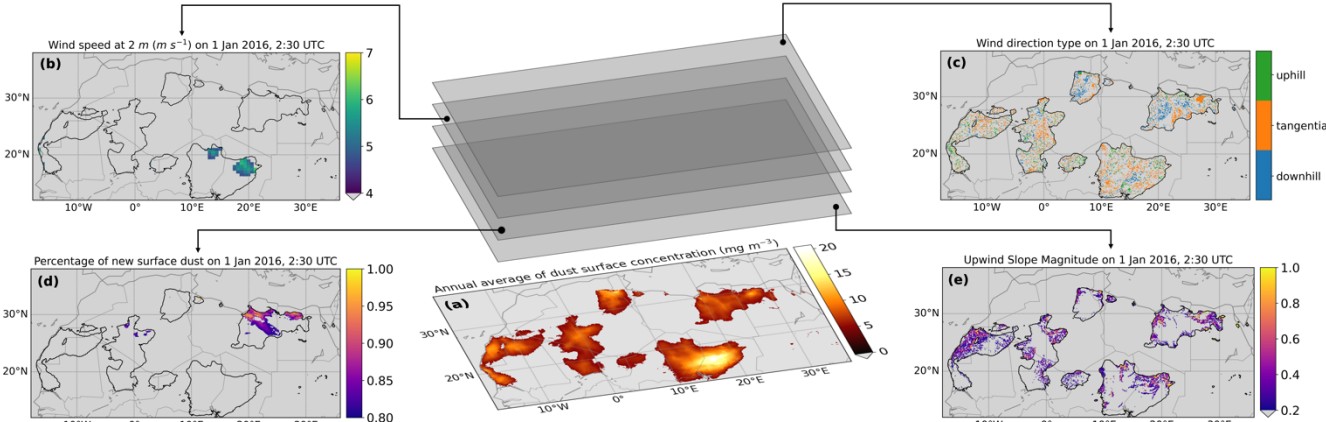

**Figure 4.** Schematics showing the procedures for screening the 10-year dust reanalysis data for fresh dust emissions. First, dust sources were
assigned to regions where the annual average of dust surface concentration is among the top percentile (i.e., 20% as shown in (a)) of all values
across the study domain of North Africa. Subsequently, additional screening criteria were applied simultaneously to events that occurred within
these dust sources. These criteria include (b) high wind speed, (c) wind direction over topography, (d) high increase in dust surface concentration,
and (e) steep slope. Maps (b–e) are examples of filters used in the "initial" run (see in Table 1) for 1 January 2016, at 2:30 UTC.

To maximize the likelihood of capturing predominantly fresh dust emissions, high wind speed was a necessary criterion, as low

wind speeds are unlikely to generate sufficient fresh dust. A single cut-off value for high wind speed was chosen as a top percentile

of the 10-year average wind speed over the whole domain. Similarly, a single threshold slope was used to select the events over

prominent terrain variations. Dust events occurring over relative flat surfaces were excluded to magnify the potential signals of

shifted PSD due to terrain variation. Wind direction over slopes was categorized following procedures described in Section 2.1

and a cut-off angle smaller than 45° was used to exclusively select typical uphill, tangential, or downhill winds over slopes. Another

criterion for increasing the probability of capturing dominantly fresh emissions was identifying sharp increases in surface

concentrations. This approach favored the initiation of significant dust emissions with relative clean background dust levels.

Examples of selected dust events based on each of the above criteria at 2:30 UTC on 1 January 2016, are provided in Fig. 4(b–e),

using the configurations for the "initial" run (i.e., a cut-off angle of 10° for categorization of wind direction, the 80[th] percentile as

threshold for high wind speeds and steep slope, and 80% as the threshold ratio of the increased surface dust concentration to the

total dust concentration as shown in Table 1).

We performed several sensitivity tests to account for the uncertainties associated with the criteria used for event selection. We

perturbed each of the threshold percentiles or the cut-off values used in the "initial" run in a pair of sensitivity runs. Specifically,

we added and subtracted 10% from the threshold percentiles of 80% used for screening dust sources, steep slope, high wind speed,

and high fraction of fresh emissions. Cut-off angles, β of 5° and 20° were tested compared to the initial 10° for typical wind

direction. Configurations of all the sensitivity runs are shown in Table 1. A combination of all the stricter criteria was used in the

"final" run.

Table 1. Thresholds or cut-off values used in all runs. Locations where the 10-year average surface dust concentration is above the threshold percentile value of all locations were designated as dust sources. Events over dust sources with slope and wind speed above the threshold percentiles of temporally averaged values over the whole domain, and increases in dust surface concentration above the threshold values were selected. The wind direction relative to topography was assigned using the cut-off angle. Values marked in bold represent the different criteria compared to the "initial" run.

| Test name | Dust sources threshold percentile | Wind direction cut-off angle | Slope threshold percentile | Wind speed threshold percentile | Increase in dust surface concentration threshold |
|---|---|---|---|---|---|
| Initial | 80 | 10° | 80 | 80 | 80% |
| 90%source | **90** | 10° | 80 | 80 | 80% |
| 70%source | **70** | 10° | 80 | 80 | 80% |
| 5cutoff | 80 | **5°** | 80 | 80 | 80% |
| 20cutoff | 80 | **20°** | 80 | 80 | 80% |
| 90%slope | 80 | 10° | **90** | 80 | 80% |
| 70%slope | 80 | 10° | **70** | 80 | 80% |
| 90%wdsp | 80 | 10° | 80 | **90** | 80% |
| 70%wdsp | 80 | 10° | 80 | **70** | 80% |
| 90%sconc | 80 | 10° | 80 | 80 | **90%** |
| 70%sconc | 80 | 10° | 80 | 80 | **70%** |
| Final | **90** | **5°** | **90** | **90** | **90%** |

## 2.4 Statistical analysis

Based on the selected events for "fresh" dust emissions, we explored the relationships between wind conditions and the PSD of emitted dust, taking into consideration the effects of various meteorological and landscape factors. Specifically, the dependent variable was the coarse fraction of surface dust, and the independent variables of focus were the wind speed and the slope associated with uphill, tangential, and downhill winds. The five additional independent variables are environmental variables that can potentially affect dust PSD as well as the relationships between dust PSD and wind conditions, including the continuous variables of year and soil moisture, and the categorical variables of time of day, season, and soil texture type. Dust events with missing values in soil texture class or wind direction type were excluded.

Exploratory data analysis was first conducted to assess data quality, identify intrinsic patterns, and guide the selection of appropriate statistical models. We selected and modified our statistical models based on their adherence to model assumptions, ability to explain the variability in the coarse fraction, and overall complexity. An initial choice was the multiple linear regression model which has the advantage of high explainability. We separated the slope by wind direction types to provide a more holistic representation of the effects of veering wind over topography. Significant coefficients for continuous variables, such as wind speed, represent the change in the coarse fraction of dust concentrations associated with a one-unit change in that corresponding independent variable. Categorical variables (e.g., time of day) are encoded as binary dummy variables, each representing a distinct category. The coefficients of these variables reflect the change in coarse fraction relative to the reference category chosen during the encoding process. Interaction terms can also be added to linear models in order to reflect the interplay between wind conditions and other factors. An interaction can be expressed as $x_m \times x_p$, where $x_m$ is one of the four wind condition variables and $x_p$ is one of the two additional continuous variables (i.e., year and soil moisture) or a dummy variable representing one of the three additional categorical variables (i.e., time of day, season, and soil texture). The adjustment in the coefficient for $x_m$ due to $x_p$ would be represented by $\beta_{mp} x_p$. A valid linear model requires linear relationships, normality of errors, constant variance (homoscedasticity), and low correlations among independent variables. Collinearity or multicollinearity among predictors can inflate standard errors and reduce the statistical significance of regression coefficients. To assess this, we calculated the Generalized Variance Inflation Factors (GVIFs) for all predictors in the linear models using the VIF function in R (R. Core Team, 2023). For categorical variables,

the GVIFs were adjusted by the degrees of freedom (Df), expressed as $GVIF^{1/(2 \cdot Df)}$. An adjusted GVIF of 1 (the smallest value) indicates no collinearity, while values below 5 generally suggest low and acceptable collinearity.

When assumptions of normality and constant variances were violated, we attempted standardization and transformations, including the Box-Cox transformation (Box and Cox, 1964) and the logit transformation (Berkson, 1944), to tackle these issues. We also tested weighted linear regression, also known as weighted least squares (WLS) (Kiers, 1997), which can handle non-constant variance (heteroscedasticity) by assigning weights to observations. In addition, Beta regression models were implemented, which are particularly useful for fractional variables that range between 0 and 1, such as the coarse fraction in this study (Douma and Weedon, 2019). Beta regression has been previously applied to air quality-related health metrics within the standard unit interval (Lu et al., 2021) and to particle size data with skewed distributions (Peleg, 2019).

Several evaluation metrics were used to compare model performance, including the root mean square error (RMSE), the mean absolute error (MAE), the adjusted coefficient of determination (adjusted $R^2$, which evaluates the amount of variability in the coarse fraction explained by the model while penalizing model complexity). Higher adjusted $R^2$, and lower RMSE or MAE values suggest that more variations of the data are captured by models. We also calculated the prediction interval accuracy at the 95% confidence level, defined as the proportion of observations covered by prediction intervals. The performance metrics were also averaged using the 10-fold cross-validation (CV), where the dataset was randomly divided into 10 subsets, and models were trained on 9 subsets and tested on the remaining subset in each iteration, with a total of 10 iterations. All statistical analyses were performed using R and the relevant packages (Zeileis and Hothorn, 2002; Grün et al., 2012; Cribari-Neto and Zeileis, 2010; Fox and Weisberg, 2019; Venables and Ripley, 2002).

Moreover, we constructed machine learning models to account for the large dataset and potential non-linear relationships. Categorical variables (wind direction, soil texture type, season, and time of day) were converted into dummy variables, resulting in a total of 22 predictors when combined with continuous predictors (wind speed, slope, and year). We built Random Forest and Extreme Gradient Boosting (XGBoost) models, both are widely used regressors (Bacanin et al., 2024; Brokamp et al., 2017; Keller & Evans, 2019; Zhang et al., 2022). Though both models rely on decision trees, Random Forest aggregates multiple trees trained on randomly sampled subset of data, whereas the XGBoost sequentially refines decision trees through iterative training. Model hyperparameters were fine-tuned to maximize the predictive performance, with the search grids determined based on sample size, predictor count, and computational efficiency. The search grids and the optimal hyperparameter combinations are listed in Table S1. As with the linear models, we assessed the accuracy of prediction interval coverage at 95% confidence level through 10-fold CV. Machine learning models are known to have high prediction accuracy but can be challenging to interpret. One technique to assess the contribution of individual predictors to the coarse fraction based on decision trees is the SHapley Additive exPlanations (SHAP) analysis, which was performed on the optimized models. These analyses were conducted using Python packages including scikit-learn (version 1.2.2; Pedregosa et al., 2011), xgboost (version 1.7.6; Chen and Guestrin, 2016), and shap (version 0.44.0; Lundberg and Lee, 2017).

Eventually, the more complex models did not outperform the multiple linear models to a large extent. Given their high explainability, ability to incorporate interactions between predictors, and competitive performance, we ultimately selected linear models for further analysis. SHAP results from the machine learning models were also included for cross-validating key findings. More details on the model performance and results are described in Section 3.3.

## 3 Results and Discussion

### 3.1 Sensitivity tests show minor variations

As a result of the screening processes, a total number of 461,183 dust events were identified in the "initial" run and 25,884 events were identified in the "final" run from around 3.5 billion records of dust surface concentrations at specific locations and times. Figure 5 shows the percentage changes in median coarse fraction by wind direction types from all sensitivity runs (Table 1). In general, nearly all perturbations of any single screening criterion result in around ±1% of changes in the average or median coarse fraction grouped by wind direction. The exception is restricting wind speed to the top 10% percentile (the "90%wdsp" run), which leads to a roughly 2% increase in the median values for each wind direction type. Coarse fraction of dust emissions with downhill winds are usually more sensitive to the screening threshold values than the other two wind directions. When all criteria were restricted simultaneously in the "final" run, the median coarse fraction of dust emission increases by less than 2% under tangential or uphill winds and around 3.5% under downhill winds. Considering the minor variations in coarse fraction among sensitivity runs, we decided to focus on one run for more detailed analysis. The "final" run was chosen because, in theory, the selected events are most representative for fresh dust emissions.

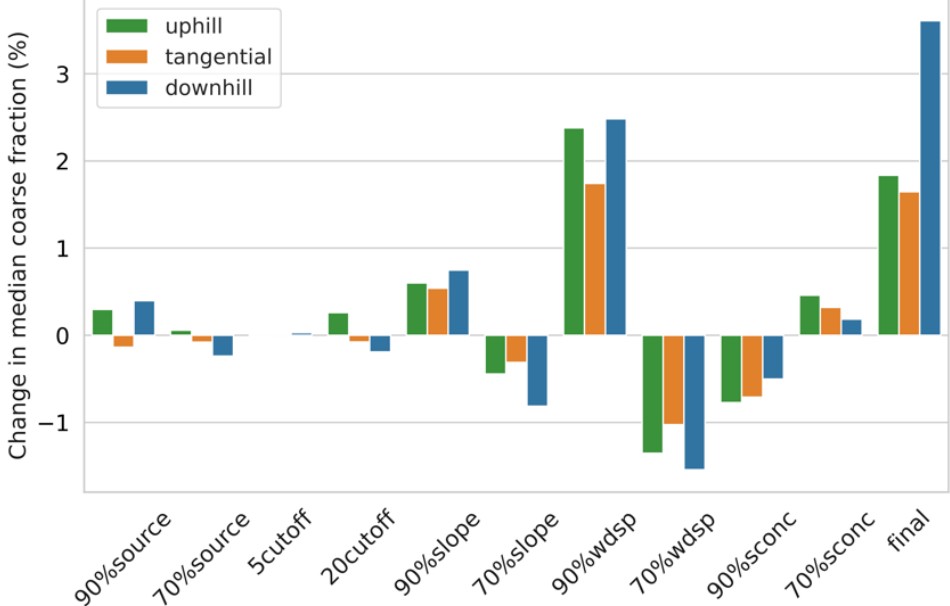

**Figure 5.** Percentage change in the median coarse fraction by wind direction type for all sensitivity runs (see Table 1) as compared to the "initial" run.

### 3.2 Exploratory analysis

To gain an impression of the general distribution of the data, we started with plotting the dust coarse fraction against four variables of wind conditions (Fig. 6). Scatter points in the panel for wind speed are color-coded by number of overlapping observations, and data points in other three panels for slopes are color-coded by wind speed. Across all four panels, a pattern of heteroscedasticity is revealed, that is, the variance of coarse fraction is greater for dust events associated with lower wind speed or slope than for events with high wind speed or slope. The vertically aligned scatter points with varying colors at a slope of around 1.7 in the panel for "slope under uphill winds" represent 93 dust events that occurred at a same location under different wind speeds near the northern border of Western Sahara during 2007–2016, illustrating how a large number of dust events can lead to highe variance. No obvious

non-linear relationships between the four wind condition variables and the coarse fraction are observed. The trend lines based on simple linear regression models of the coarse fraction against each wind condition variable indicate general trends in the dust PSD with varying wind conditions, but the significance of these relationships is not assured. Additional plots for the soil moisture and slope under three wind conditions color-coded by the density of overlapping data points are presented in Fig. S2.

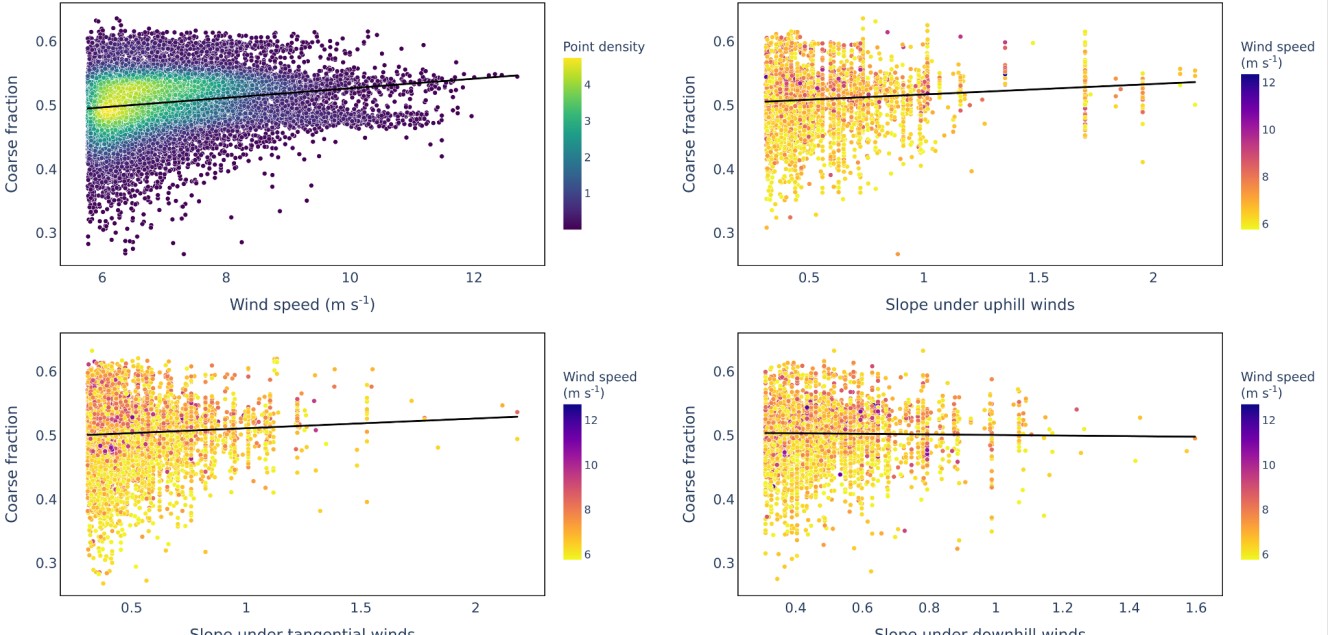

**Figure 6.** Scatter plots and linear trend lines of relationships between the coarse fraction of surface dust concentration in fresh emissions and the wind speed and the slope under three different wind directions. In the upper left panel for wind speed, the color-codes present the number of overlapping data points. In the other three panels for slope, the color of each scatter point represents the associated wind speed for each dust event.

### 3.3 General associations between wind conditions and dust coarse fraction

The absence of obvious non-linear relationships between coarse fraction and wind conditions from the exploratory analysis further motivated us to initially construct linear regression models, in addition to it being a common starting point. The linear model including all independent variables but no interactions has an adjusted $R^2$ value of 0.224, RMSE with 10-fold CV of 0.070, and MAE with 10-fold CV of 0.059. Additionally, we added all possible interactions into the model and fine-tuned it by removing the insignificant interaction terms. The resulting model with interactions has similar performance, with the adjusted $R^2$ of 0.239, RMSE with 10-fold CV of 0.070, and MAE with 10-fold CV of 0.058. The somewhat weak correlations may be related to factors that are not included in the model, such as deposition, variability within the same categories of soil texture, etc. Residual analyses indicate violations of the assumptions of evenly distributed variance and normality (Fig. S3). We attempted to address these issues and improve model performance through various linear model adaptations but only obtain indefinite or marginal improvements—standardizing the coarse fraction and wind conditions variables yields an adjusted $R^2$ of 0.241, RMSE with 10-fold CV of 1.678, and MAE with 10-fold CV of 1.420; and logit transformation on the coarse fraction generates an adjusted $R^2$ of 0.237, RMSE with 10-fold CV of 1.420, and MAE with 10-fold CV of 0.487. Furthermore, the Box-Cox transformation, weighted least squares (WLS), or Beta regression models with the best-performing configuration (with a log-log link function for the mean and an identity

link function for the dispersion) all fail to resolve heteroscedasticity in the residuals with respect to wind condition variables. For the machine learning models, the optimized Random Forest and XGBoost models achieve R² values of 0.407 and 0.474, respectively, which drop to 0.259 and 0.273 after 10-fold CV, indicating potential overfitting. The coverage rates of prediction intervals with the 10-fold CV are 92.5% for Random Forest and 54.4% for XGBoost, both lower than the 94.0% coverage by the linear models both with and without interactions. Linear models outperforming machine learning models on prediction accuracy strongly encourages the selection of linear models. Considering their comparative satisfactory performance, simplicity, and directly interpretable coefficients, we decided to proceed with the linear regression models among all models. We first assessed the influence of individual predictors based on the linear regression model without interactions, with cross comparison with results from the machine learning models (Fig. S5). Subsequently, we used the linear model with interaction terms to investigate the effects of interactions among predictors.

Linear model without interaction terms are used to infer the general effects of wind conditions on coarse dust fraction. These linear models are not intended to imply strictly linear relationships between dust PSD and wind conditions, but rather to provide initial guidance on the directionality of these relationships. Although individual data points present deviations, our models effectively predict the overall trend, as suggested by the response vs. fitted value plots (Fig. S4), where the predicted values align closely with observed values and cluster around the one-to-one red line. Adjusted GVIF values for the model without regression were consistently below 2 with most values close to 1 (Table S2), indicating that multicollinearity among continuous or categorical predictors does not significantly affect the regression model. The multiple linear regression model for the dust coarse fraction includes four independent variables for wind conditions (speed and three options for slope) and additional factors that may affect the PSD of dust emissions, allowing us to investigate the effects of topographic wind conditions while controlling interferences from other environmental factors. The model can be expressed as:

$$y = \beta_0 + \beta_1 x_1 + \beta_2 x_2 + \beta_3 x_3 + \beta_4 x_4 + \beta_5 x_5 + \beta_6 x_6 + \beta_7 x_7 + \beta_8 x_8 + \beta_9 x_9 + \epsilon, \tag{3}$$

where, $y$ represents the coarse fraction of dust emissions, $x_1$ represents wind speed, and $x_2$, $x_3$, and $x_4$ represent slope under uphill, tangential, and downhill winds, respectively; $x_5$ is the categorical variable of time of day, including three levels of morning (6:00–12:00 local time), afternoon (12:00–18:00 local time), and evening (18:00–6:00 local time); $x_6$ is the categorical variable of season, comprising DJF (winter months of December, January, and February), MAM (spring months of March, April, and May), JJA (summer months of June, July, and August), and SON (autumn months of September, October, and November); $x_7$ and $x_8$ represent the continuous variables of year and soil moisture; and $x_9$ is the categorical variable of soil texture class, which contains eight levels of the FAO soil texture classes (Jahn et al., 2006). The coefficients $\beta_i$ represent the expected changes in the response variable $y$ per unit increase in the continuous predictor $x_i$, and the difference in $y$ relative to the reference category for categorical predictor $x_i$, while holding all other variables constant. The $\epsilon$ represents the residuals of the model. The default coarse fraction is cf2 (defined in Section 2.1), and the corresponding estimated values, standard errors, and significance of the wind condition coefficients ($\beta_1 - \beta_4$) are presented in Table 2, with full details of all coefficients in Table S3. As noted in Section 2.1, we also tested two alternative definitions of coarse fraction (cf1 and cf3) and compared the estimated coefficients with their statistical significance in Table S4. The coefficient estimates based on cf2 and cf3 are largely consistent, whereas those based on cf1 show some distinct patterns. Given that dust in the top bin (12-20 μm) falls into the "super coarse" dust category (> 10 μm; Meng et al., 2022), these results

suggest that super coarse dust responds differently to varying wind conditions compared to other coarse dust. Therefore, the cf2 definition serves as a robust representation of coarse dust particles.

The coefficient for wind speed from the linear model, $\beta_1$ is statistically significant and positive (Table 2), suggesting that controlling the effects of other factors, the coarse fraction of dust emissions increases with the wind speed. This result is consistent with the SHAP analyses on both machine learning models as showcased in the SHAP summary plots (Fig. S5), where purple data points with high wind speed are concentrated on the right side of the vertical centerline and yellow points with lower wind speed dominate the left side, indicating a positive correlation between wind speed and coarse fraction. The observed trend contradicts implications of the theory based on the saltation-bombardment emission mechanism, which predicts that higher kinetic energy of impact particles from greater wind speed can intensify the disintegration of soil aggregates and thus the release of finer particles (Shao, 2001; Alfaro et al., 1997). Conversely, our result aligns with the observed shift in the dominant emission mechanism from "shaking-off" of submicron particles to the generation of coarser microparticles from fragmentation as the velocity of saltating particles increases (Malinovskaya et al., 2021). An alternative explanation for the observed positive effect of wind speed on dust size at emission is related to soil conditions (Ishizuka et al., 2008; Panebianco et al., 2023). In previous studies, emissions of super coarse dust (> 10 μm) increased with wind speed while the emissions of ultrafine dust (< 1 μm) remained nearly invariant over sandy soil (Panebianco et al., 2023); in addition, the fraction of fine dust (< 2 μm) decreased with friction velocity on slightly crusted surfaces (Ishizuka et al., 2008). These phenomena were likely due to weaker cohesive forces and thus easier emission of coarse particles than fine particles. Though our sensitivity test using cf1 rejects the increased emission of super coarse dust with wind speed (Table S4), unmeasured changes in the fine dust emissions leading to an overall higher coarse fraction remain one possibility. Soil texture and soil moisture were included in the model, but subtle discrepancies across events within the same soil class or soil moisture are not eliminated. Last but not least, since the fresh dust emissions at regional scale inevitably include transported dust, yet another potential explanation is unrelated to the emission, but to the transport process—as wind speed increases, more fresh emissions are generated, which undergo less deposition and contain a higher fraction of coarse particles than the aged background dust (González-Flórez et al., 2023).

**Table 2.** Estimates, standard errors, and p-values of wind condition coefficients for the multiple linear model of dust coarse fraction. The model includes the independent variables of wind conditions (i.e., wind speed and slope under three wind direction types), time of day, season, year, soil moisture, and soil texture. The symbols of coefficients are defined in Eq. (3).

| Coefficients for Variables | Estimates | Standard errors | p-values |
|---|---|---|---|
| wind speed ($\beta_1$) | 0.0075 | 0.0002 | <0.0001 |
| slope with uphill winds ($\beta_2$) | 0.0175 | 0.0013 | <0.0001 |
| slope with tangential winds ($\beta_3$) | 0.0081 | 0.0015 | <0.0001 |
| slope with downhill winds ($\beta_4$) | 0.0076 | 0.0016 | <0.0001 |

The coefficients for slopes under all three wind directions from the linear regression are significant and positive (Table 2), suggesting that the coarse fraction of dust emissions increases with the slope regardless of the relative wind direction. The largest coefficient for uphill winds among all three slopes indicates that it has the strongest effect on dust coarse fraction. Similarly, the SHAP analysis on the optimized Random Forest model (Fig. S5) suggests a positive relationship between slope and coarse fraction, with uphill winds further accentuate the positive effects. The XGBoost model indicates mixed effects of wind directions but the results are less reliable due to its lower prediction accuracy (54%). Overall, the linear model and the Random Forest model agree on the positive correlation between slope and dust size, especially under uphill winds. The strong increase in coarse fraction with uphill slope aligns with previous findings using large eddy simulations, which was explained by the enhancement in vertical

transport of dust particles being more prominent for coarser particles (Heisel et al., 2021). In contrast, the microphysics of dust emission proposed that compared to tangential winds, uphill winds against the slope resulted in more detachment of fine particles from the surface of soil aggregates on the windward slope due to the secondary aeolian structures, and meanwhile less ejection of coarse particles from the fragmentation of soil aggregates upon hitting the leeward slope (Malinovskaya et al., 2021). Our results suggest that at the regional scale, the effect of near-source transport of emitted dust at scales of hundreds to thousands of meters dominates over the impact of microphysics of dust emission related to secondary dune structure at scales of centimeter to meters. The overall elevated coarse fraction with slopes might also be attributed to the orographic wind channelling (Rosenberg et al., 2014), increased availability of coarser particles on hills (Samuel-Rosa et al., 2013; Washington et al., 2006) and their greater mobility under gravity. Effects of slope under tangential and downhill winds are less pronounced, with linear model suggesting their weaker positive relationships with coarse fraction compared to uphill winds (Table 2), and the Random Forest model indicating negative impacts of tangential and downhill wind directions on coarse fraction even though the effect of slope is positive (Fig. S5). These weaker effects can possibly be explained by the lack of effective enhancement in vertical transport of coarse particles on the windward side of slopes.

**3.4 Associations between wind conditions and dust coarse fraction under varying environmental conditions**

Adding interaction terms to the linear model allow us to investigate how the relationships between wind conditions and dust coarse fraction may vary depending on the additional variables for time and surface characteristics. The model including all significant interactions is shown in Eq. (4). Results for coefficients related to the interactions are shown in Table 3 and the complete results are included in Table S5.

$$y = \beta_0 + \beta_1 x_1 + \beta_2 x_2 + \beta_3 x_3 + \beta_4 x_4 + \beta_5 x_5 + \beta_6 x_6 + \beta_7 x_7 + \beta_8 x_8 + \beta_9 x_9 + \beta_{15} x_1 x_5 + \beta_{16} x_1 x_6 + \beta_{18} x_1 x_8 + \beta_{25} x_2 x_5 + \beta_{26} x_2 x_6 + \beta_{28} x_2 x_8 + \beta_{35} x_3 x_5 + \beta_{38} x_3 x_8 + \beta_{46} x_4 x_6 + \epsilon, \tag{4}$$

where, $x_1 - x_9$, $y$, and $\beta_1 - \beta_9$ are the same as in Eq. (3). The $\beta_{ij}$ are coefficients for interactions between $x_i$ ($1 \le i \le 4$) and $x_j$ ($5 \le j \le 9$) and their interpretations are described in Section 2.4.

With interactions in the model, the coefficient $\beta_1$ indicates the slope of linear correlation between wind speed and the coarse fraction when the variables that have interactions with wind speed are at the reference levels for categorical variables (i.e., "afternoon" for time of day and "DJF" for season) or at zero for continuous variable (i.e., soil moisture). Adjustments of the correlation under other conditions are indicated by the coefficients for interactions with wind speed ($\beta_{15}$ $\beta_{16}$, and $\beta_{18}$). The overall coefficient for wind speed stays positive with varying time of day and season, which agrees with the results from the model without interactions (Table 2), except for the rare cases when soil moisture is high ($> 50\%$). As suggested by the adjustments of coefficients, the positive correlation between wind speed and coarse fraction is weakened during events that happen in the afternoon, in summer, or are associated with higher soil moisture. All these conditions are typical for haboob dust storms which are capable of generating intense dust emissions (Heinold et al., 2013; Knippertz, 2017). Therefore, a potential explanation for the observed patterns is that the dust PSD dependency on wind speed is reduced during convective conditions associated with haboobs. Reasons behind the weakened correlation could be related to turbulent atmospheric conditions. If the earlier assumption is valid that the coarse fraction increases with wind speed due to the associated higher proportion of fresh emissions, during convective events, the role of turbulent flows in keeping dust particles suspended regardless of the magnitude of wind speed may blur the effects of wind speed. Moreover,

the positive relationship between wind speed and coarse fraction diminishes with soil moisture. Higher soil moisture may inhibit fresh dust emissions, thereby weakening the positive correlation.

Table 3. Estimates, standard errors, and p-values of the interaction coefficients for the multiple linear model of dust coarse fraction. The model includes the independent variables of wind conditions (i.e., wind speed and slope under three wind direction types), time of day, season, year, soil moisture, and soil texture, as well as significant interaction terms between wind conditions and other independent variables. The interaction coefficients represent wind conditions (speed and direction) under various situations of time of day, season, and soil moisture. The symbols of coefficients are defined in Eq. (3) and (4). Statistically significant (at 0.05 significance level) coefficients are bolded and their p-values are marked with "*", among which the negative coefficients are italicized.

| | Estimates | Standard errors | p-values |
|---|---|---|---|
| **Multiple linear model coefficients for wind speed under various conditions** | | | |
| Afternoon, DJF, and soil moisture of 0 (reference levels; $\beta_1$) | **0.0076** | **0.0007** | **<0.0001*** |
| Adjustments with time of day ($\beta_{15}$) | | | |
| evening | **0.0122** | **0.0006** | **<0.0001*** |
| morning | **0.0016** | **0.0006** | **0.0058*** |
| Adjustments with season ($\beta_{16}$) | | | |
| JJA | *-0.0028* | *0.0007* | *<0.0001** |
| MAM | *-0.0023* | *0.0006* | *0.0003** |
| SON | -0.0003 | 0.0007 | 0.6500 |
| Adjustments with soil moisture ($\beta_{18}$) | *-0.0154* | *0.0029* | *<0.0001** |
| **Multiple linear model coefficients for slope with uphill winds under various conditions** | | | |
| Afternoon, DJF, and soil moisture of 0 (reference levels; $\beta_2$) | **0.0135** | **0.0030** | **<0.0001*** |
| Adjustments with time of day ($\beta_{25}$) | | | |
| evening | **0.0061** | **0.0024** | **0.0118*** |
| morning | **0.0159** | **0.0026** | **<0.0001*** |
| Adjustments with season ($\beta_{26}$) | | | |
| JJA | *-0.0098* | *0.0028* | *0.0005** |
| MAM | *-0.0138* | *0.0029* | *<0.0001** |
| SON | -0.0056 | 0.0031 | 0.0672 |
| Adjustments with soil moisture ($\beta_{28}$) | 0.0521 | 0.0107 | <0.0001 |
| **Multiple linear model coefficients for slope with tangential winds under various conditions** | | | |
| Afternoon, soil moisture of 0 (reference levels; $\beta_3$) | -0.0038 | 0.0025 | 0.1261 |
| Adjustments with time of day ($\beta_{35}$) | | | |
| evening | **0.0134** | **0.0024** | **<0.0001*** |
| morning | **0.0110** | **0.0027** | **<0.0001*** |
| Adjustments with soil moisture ($\beta_{38}$) | **0.0351** | **0.0115** | **0.0022*** |
| **Multiple linear model coefficients for slope with downhill winds under various conditions** | | | |
| DJF (reference level; $\beta_4$) | **0.0148** | **0.0026** | **<0.0001*** |
| Adjustments with season ($\beta_{46}$) | | | |
| JJA | *-0.0101* | *0.0031* | *0.0011** |
| MAM | *-0.0105* | *0.0032* | *0.0011** |
| SON | *-0.0090* | *0.0036* | *0.0116** |

The adjustments of the relationship between slope and coarse fraction with other environmental variables are generally consistent across three wind direction types. Overall, the coefficients for slope with three wind directions stay positive under most

circumstances, aligning with results from the model without interactions (Table 2). Notably, the effect of the uphill slope is strongest in the morning. Unlike Harmattan surges, which can induce dust emission throughout the day, or haboob storms, which mostly occur in the afternoons, dust uplift due to the breakdown of night-time low-level jets (NLLJs) is limited to the period around sunrise to midday (Fiedler et al., 2015; Heinold et al., 2013). Therefore, the result may indicate that the role of uphill slope in facilitating transport of coarse dust is particularly relevant during emissions related to NLLJs. Moreover, the effect of slope in increasing coarse fraction is weakest during afternoon events under both uphill and tangential winds. With both uphill and downhill winds, the positive correlation between slope and coarse fraction is the strongest in winter, and the weakest in spring and summer. The reduced correlation of dust PSD with uphill slope in both afternoon and summer suggests a diminished effect of slope during haboob dust storms. This can be explained by the stronger turbulence associated with convective storms, which readily stirs up the air and facilitate particle transport, thereby weakening the additional effect of uphill slope by elevating coarse particles through flow separation.

The effect of slope on increasing coarse fraction of dust also becomes more apparent with increasing soil moisture under uphill and tangential winds. A potential explanation is that low soil moisture might be associated with low water vapor content in lower-Saharan Air Layer, which can lead to continued vertical motions of the atmosphere into the night due to increased atmospheric longwave heating (Ryder, 2021). Conversely, the air is more stable with higher relative humidity, making the enhancement by topography more critical for the transport of coarse dust.

## 4 Conclusion

This study aims to explore the relationship between topographic wind conditions and particle size distribution (PSD) of dust emissions on a regional scale through data analysis. The Multiscale Online Nonhydrostatic AtmospheRe Chemistry model (MONARCH) dust concentrations were first evaluated against flight measurements of fresh dust emissions from the 2011 Fennec campaign and were proven to be effective in capturing concentrations of coarse to super coarse dusts in fresh dust emissions. For our analysis, size-resolved surface dust mass concentrations from the MONARCH dust reanalysis over the Sahara during 2007–2016 were condensed into an index of coarse fraction (the ratio of the sum of concentrations in the top two bins (6–20 µm) to the total concentration in eight bins (0.2–20 µm)), serving as the proxy for size distribution. Information on wind vectors and soil moisture, elevation, and soil texture was obtained from the Modern-Era Retrospective analysis for Research and Applications (MERRA-2) reanalysis data, the NASA Shuttle Radar Topography Mission Global 3 arc-second (SRTM GL3) dataset, and the inputs to the Global Land Data Assimilation System version 2 (GLDAS2) Noah land surface model, respectively. Several highly selective criteria were applied to maximize the probability of selecting fresh dust emissions with typical wind conditions over topography. Scatter plots of coarse fraction against four wind conditions variables (i.e., wind speed, slope under uphill, tangential, and downhill winds) reveal unevenly distributed variance without obvious non-linear trends. We ultimately selected the multiple linear models after testing several model variations to quantify and explain the trends in data, with key findings cross-validated using machine learning models.

The linear model without and with significant interaction terms can explain 22% and 24% of the variability of coarse fraction, respectively. The model, however, fails to fulfil the assumptions on homoscedasticity (constant variance) and normality, and this issue could not be resolved by other parametric modelling approaches including linear regression models with transformations of variables, weighted least square, or Beta regression models with several options for link functions. The strong intrinsic pattern of

non-constant variance and the abundance of data points require more advanced models, which is beyond the scope of our current work. Other uncertainties arose from the varying original resolution of datasets and the resampling process. Moreover, even though we applied multiple criteria to exclusively pick fresh dust emissions, we cannot totally exclude the portion of transported dust. The analysis focuses on the general trend for North Africa, and more detailed insights rely on analysis for smaller geographic domains.

Despite some limitations, the multiple linear models achieved high predictive accuracy—over 94% under 10-fold cross-validation (CV)—demonstrating their capability to provide meaningful insights for interpretation. The optimized Random Forest model, which attained 92% predictive accuracy with 10-fold CV despite potential overfitting, added insights to the influence of each predictor by applying the SHapley Additive exPlanations (SHAP) analysis. Both the linear and the Random Forest models reveal positive associations between coarse dust fraction and both wind speed and slope. An increased coarse fraction in dust emissions with wind speed is inconsistent with theories by Kok (2011) or those based on saltation-bombardment mechanism (Shao, 2001; Alfaro et al., 1997), which predict invariant or opposite trends, respectively. The most likely explanation for our finding is that higher winds are associated with more fresh emissions (which undergo less deposition) because transported dust was inevitably included in our samples even with meticulous screening (González-Flórez et al., 2023). Therefore, our finding does not necessarily reject the former theories, as the definition of dust emissions may differ across studies. Another notable possibility is a shift in emission mechanisms with wind speed—under light winds, the detachment of fine particles from surfaces of soil aggregates dominates the emissions, generating more fine dust; in contrast, the fragmentation of soil aggregates dominates emissions under stronger winds and ejects more coarse dusts (Malinovskaya et al., 2021).

The positive correlation between coarse dust fraction and slope is most pronounced under uphill winds. This aligns with the impacts of flow separation induced by topography, as suggested by previous numerical simulations (Heisel et al., 2021), but contradicts the observed effects of frontal winds explained by the microphysics of dust emissions (Malinovskaya et al., 2021). Our finding suggests that the topographic influence on transport over hundreds to thousands of meters may override the localized effects (increased generation of fine particles) from secondary aeolian structures at scales of centimeters to meters. The persistent influence of slope on increasing the coarse fraction, regardless of wind direction, might be related to meteorological conditions (e.g., orographic wind channelling; Rosenburg et al., 2014) and soil conditions (e.g., increased availability and gravitational mobility of coarser particles on hills; Washington et al., 2006).

Including interaction terms in the model allows us to investigate shifts in the effects of wind conditions on dust size under different environmental conditions. The positive correlation between wind speed and coarse fraction diminishes during afternoon and summer events and under high soil moisture. This reduction is likely due to decreased differences in dust size distribution by deposition during haboob convective storms when turbulence is strong (Heinold et al., 2013; Knippertz, 2017). The uphill slope exhibits the strongest effect on increasing dust size in the morning, suggesting that the enhanced vertical transport may be particularly effective in uplifting coarse dust during emissions related to the breakdown of night-time low-level jets (Fiedler et al., 2015; Heinold et al., 2013). The effect of uphill slope is weakened during summer and afternoons, indicating that turbulence during haboob dust storms has competing effects in sustaining airborne coarse dust.

This work provides insights into the controlling factors of dust PSD on a regional scale using a meta-analysis of a 10-year dust reanalysis dataset, complementing the accumulating knowledge from recent field measurements. The study highlights the overlapping effects and interactions among various environmental factors on the size distribution of dust emissions, which can potentially be applied to improve dust emission parameterizations in regional to global Earth system models. Moreover, the workflow for screening fresh dust events developed in this work serves as a reference for future studies utilizing datasets at various scales. Remaining uncertainties in this work (e.g., those introduced by unmatching resolution of source data or lack of explicit

information on dust event types) calls for further investigation, especially on the role of various environmental factors and their interactions. Additional field measurements, as well as the development and validation of data products providing detailed information on size-resolved dust emissions, meteorological conditions, and soil and topographic properties, would be crucial for advancing this field.

**Data Availability**

The open-access MONARCH dust reanalysis data, prepared by the Barcelona Supercomputing Center (BSC), are available at https://earth.bsc.es/thredds_dustclim/homepage/. The Modern-Era Retrospective analysis for Research and Applications (MERRA-2) data (Gelaro et al., 2017) are managed by the NASA Goddard Earth Sciences (GES) Data and Information Services Center (DISC) and can be accessed at https://disc.gsfc.nasa.gov/datasets?project=MERRA-2. The NASA Shuttle Radar Topography Mission Global 3 arc-second (SRTM GL3) dataset is available at https://lpdaac.usgs.gov/products/srtmgl3v003/. The soil texture map input to the Global Land Data Assimilation System version 2 (GLDAS2) Noah land surface model (Rodell et al., 2004) is described and provided by NASA at https://ldas.gsfc.nasa.gov/gldas/soils.

**Competing Interests**

The authors declare that they have no conflict of interests.

**Author Contribution**

HF secured funding for the research. HF and XH conceptualized the study, developed methodology for data collection and processing, and provided the scientific interpretation of the results. XH processed the data and created the data visualizations. WG conducted the statistical analysis of the processed data, drafted the corresponding methods (Section 2.4), and verified the initial interpretation based on the statistical results. XH wrote the initial draft of the manuscript, and all authors contributed to the final version.

**Acknowledgements**

This work was supported by the National Aeronautics and Space Administration (NASA) under Grant 80NSSC20K1532. We are grateful to Dr. Clair Ryder for her guidance on the Fennec campaign data, and to Dr. Carlos Pérez García-Pando for reviewing the manuscript and providing valuable insights into the interpretation of the results.

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
