# Peer review of "Impact of Topographic Wind Conditions on Dust Particle Size Distribution: Insights from a Regional Dust Reanalysis Dataset"

_EGUsphere, 2024_

## Referee Comment (RC1)

This article analyzes the association between the particle size distribution of windblown dust and topographical wind conditions over the Sahara, using linear regression models with inputs from the coarse fraction of dust from MONARCH dust reanalysis dataset, the wind conditions from MERRA2 meteorological reanalysis fields, and surface elevation data. Positive correlations between particle size and wind speed and uphill slope wind direction are found in this study. The scope of the manuscript is very important. However, I am concerned about the soundness of the method and thus the associated interpretation and conclusions.

**General Comments**

1.  The linear regression model used in the study to state the relationship could be misleading and lack of strong evidence. The dust emissions are inherently nonlinear and vary with the cube (subjected to land surface properties) of surface friction velocity. The initiation of dust emissions is also subjected to the threshold friction velocity. The application of linear regression model and simple treatment of adding interaction terms between independent variables impose violation against the nonlinear processes in dust emissions, transport and deposition.
2.  The poor explainability from the linear regression model without interaction terms ($R^2$ of 0.224) and with interaction terms ($R^2$ of 0.239) between independent variables questions the soundness of the results and interpretability. Thus, the interpretation from the manuscript is not based on strong evidence and at a worse case potentially causes misleading conclusions.
3.  The coarse resolution of input data and the validity of capturing fine-scale terrain-induced wind fields and dust emissions are not strongly evaluated. The coarse resolution of MERRA2 at $0.5° \times 0.625°$ cannot resolve localized wind fields over regions with steep slopes, and the usage of 2-m winds from MERRA2 cannot represent actual localized 2-m winds due to elevation averaging. The manuscript is heavily based on the MONARCH dust reanalysis dataset. Although it has satellite assimilation embedded and shows generally agreement against regional mean measurements, it is still questionable to resolve the fine-scale dust emissions or concentrations. Considering the target of this manuscript over locations with prominent surface elevation changes, fine-scale variability is especially important to gain insights. Thus, the coarse resolution of independent variables, and questionable fine-scale validity of dependent dust concentrations from MONARCH can impose severe reliability of the interpretation and conclusions from this manuscript.
4.  As said in the paper, MONARCH assimilates coarse dust optical depth (DOD) from satellite with fixed first-guess particle size distribution of emitted dust. How would the uncertainties for the assimilation of coarse DOD propagate into the particle size distribution of dust?

5. The article uses coarse fraction of dust concentrations as a surrogate for the particle size distribution. This is an importance piece of information. I recommend clarifying it in the abstract.

**Specific Comments**

1. Line 101-103: what is the performance of first-guess particle size distribution of emitted dust compared to Fennec?
2. Line 109-111: How would the interpretation be sensitive to the definition of coarse fraction used here? For example, how would the results change when using coarse fraction as the mass of largest dust bin over the total mass of dust?
3. Line 186-189: From Figure 3, it looks like the size distribution for size bins of 0.6-12 $\mu$m is overestimated by MONARCH compared to Fennec. How would that affect the analyses for particle size distribution of dust?
4. For Figure 6, the points overlap with each other too much, making it hard to see clearly. Could it help to show the results with the number of points color coded?

---

## Referee Comment (RC2)

General comments:

The manuscript titled "*Impact of Topographic Wind Conditions on Dust Particle Size Distribution: Insights from a Regional Dust Reanalysis Dataset*" by Huang et al. is a manuscript that attempts to investigate what factors change the emitted dust particle size distribution (PSD). They did this analysis by employing the MONARCH chemical transport model with the assimilated dust optical depth (DOD). The authors claimed that by assimilating MONARCH dust using observed DOD, MONARCH has the ability to produce adequate emitted dust PSD. The authors used the MONARCH surface dust concentrations to construct atmospheric dust concentration PSD, claiming this represent the emitted dust PSD. The authors tried to truncate aged transport dust and retain fresh dust by selecting only strongly and freshly emitting dust events using several criteria. Then, the authors performed a multiple linear regression (MLR) analysis to quantify the sensitivity of coarse dust fraction (coarse dust concentration divided by total dust concentration) to various meteorological (wind speed, wind direction) and land-surface variables (e.g., topography, soil texture, soil moisture). The authors found out a predictor model that shows that coarse dust fraction increases with wind speed and with hillslope. They also showed that some times of the day or some seasons favor the production of coarse dust.

This is an interesting and impactful study showing that the observed coarse dust might not be decreasing with or invariant with wind speed as some of the previous theories depicted (e.g., Shao, 2001; Kok, 2011). Therefore, the contribution of this paper is significant. The paper is also nicely written. However, since the paper is based on model/reanalysis and not observations, the authors need to show enough evidence to convince us that MONARCH produces correct dust PSDs. I have some major and minor comments below concerning some parts of the arguments and analysis, and I suggest a major revision.

Major Comments:

1a. What I concern the most is that the findings from this paper are highly contingent upon how successfully MONARCH assimilates the fresh dust concentration PSD. All major findings of this paper are based not on observations but on the MONARCH reanalysis data. As the authors said, the correlation between coarse dust 'concentration' fraction and wind speed can be due to other reasons (e.g., turbulence can hold coarse dust longer in the atmosphere). Even though the authors tried to focus on fresh dust events to minimize the transport effect, the correlation could still be just because there are just more fresh dust particles that contains coarse dust, as the authors referred to Cristina's argument several times (Gonzales-Florez et al., 2023). This means that the correlation the authors found will only be valid for atmospheric dust concentration PSD, but not for dust emission PSD.

1b. Following the previous point, it is great that MONARCH shows this correlation between coarse dust concentration fraction and wind speed, but this is not only the evidence in concentration (not in emission) but also model evidence (not from in-situ measurements). So, the current evidence to me is not strong enough to reject Shao's saltation bombardment theory (Shao, 2001) or Kok's brittle fragmentation theory (Kok, 2011), which focused on the dust 'emission' PSD. But, at least, if MONARCH' dust PSD is correct, then other chemical transport models using Shao or Kok's theories should still replicate this correlation between coarse dust concentration and wind speed.

2. Line 103: Why do you think that MONARCH, by only assimilating coarse-mode DOD (and not fine-mode DOD), is enough to fully constrain the whole dust PSD? (I surely know we can hardly isolate fine dust from other fine-mode aerosols in MODIS). The PSD curve has two ends, and MONARCH

only assimilates coarse-mode DOD, which constrains the right/high end. There is no information to constrain the left/low end, and MONARCH could be underestimating fine dust by not assimilating fine-mode dust. The correlation between coarse dust and wind speed might be a spurious artifact from MONARCH.

3. Lines 106-107: Following the first point, dust surface concentration PSD is not dust emission PSD, although they tend to correlate more in fresh dust events. It inevitably includes also transported dust, and so using dust concentrations PSD already underestimates coarse dust particles. Does this partially weaken the slope and $R^2$ you found?

4. Lines 176-180: I am curious about Claire Ryder's FENNEC data. Since they have several FENNEC flights, did the authors see a correlation between the coarse dust fraction from her fresh dust PSD and wind speed? It would be better to look at observed correlations too.

5a. Line 189: Since the whole paper's findings is contingent upon the accuracy of MONARCH's assimilated dust concentration PSD, I am not sure if one plot (Fig. 3) of MONARCH vs FENNEC is enough for model evaluation. Fig. 3 only shows that MONARCH's simulating adequate coarse dust, but it does not show that the posterior (assimilated PSD) is doing better than the prior (first guess PSD), and it is hard to judge from Fig. S1. Could you include the first guess in Fig. 3 in a different color and describe how the assimilated PSD is better than the first guess?

5b. Moreover, this evaluation in Fig. 3 does not show that the correlation between coarse dust fraction and wind will hold true in observations.

6. Lines 115-117: I am concerned that this paper would yield very different regression results if the authors used MONARCH's predictor variables rather than MERRA-2's fields. Please describe how MONARCH has its own driving meteorology. Did they do any meteorological assimilation or was it a complete free run? Please also briefly describe how we should expect MONARCH's met fields to be different from MERRA-2's met fields, and the implications on your regression analysis (e.g., will the sensitivities of coarse dust to wind directions largely enhance using MONARCH's meteorology?)

7a. Line 255-257: Since year, season, and time of day are correlated with winds and soil moisture, adding them in your regression analysis likely weakens the $R^2$ of winds and soil moisture on dust PSD. Please comment on how this collinearity between predictor variables impacts the regression results.

7b. Following the previous comment, wind speed and soil moisture are highly correlated. Including soil moisture likely highly changes the value of $\beta_1$.

8. Lines 313: I agree with another reviewer that the $R^2$ of the regression analysis is rather low. From Fig. 6, it looks like to me that instead of a spatial regression analysis, it might be better construct regression models for each grid/small region rather than putting them together into a single plot.

9. Lines 364-365: This statement does not sound correct to me. A multiple linear regression (MLR) analysis does not give the slope/sensitivity of a predictor variable while holding other predictor variables constant, i.e., it is not a partial differential $\frac{\partial(\text{coarse dust})}{\partial(\text{slope})}$. The magnitude of the slope from MLR is heavily impacted by the multicollinearity among different predictor variables. It is all right to say that coarse dust increases with slope regardless of wind direction, but other factors are not held constant in the MLR when this slope is estimated. Please clarify this sentence.

10. Lines 372-376: The model shows that regardless of wind direction (uphill, tangential, downhill), coarse dust fraction increases with slope. Normally we will expect if coarse dust increases with more uphill slope, then the opposite should be true (fine dust increases with more downhill slope). The three explanations to why increasingly uphill, tangential, and downhill slopes can all generate more coarse dust fraction sound a little ad hoc. Under what circumstance should the fine particle fraction increase?

Other comments:

Lines 72-74: Does this finding come from Malinovskaya 2021? Please cite if so. Does this conflict this paper's findings?

Lines 97-99: A reference is needed on how coarse-mode DOD was obtained from Aqua/MODIS, so people do not need to go to MONARCH's paper to look for it. Is it Paul Ginoux's method?

Lines 100-102: What prior (first guess) dust PSD does MONARCH assume? This requires a more detailed description here. How much is the posterior dust PSD (after assimilation) depend on the assumed prior dust PSD? This question also needs some elaboration.

Line 103: Did MONARCH "nudge" the size-resolved concentrations or used a Kalman filter? They are different methods.

Lines 129-131: A description of how MERRA-2's assumed topography can alter MERRA-2's winds is needed. MERRA-2 has a coarse grid resolution of $0.5°$, so the winds can only see a regional topographic slope. Also, I think that SRTM GL3's topographic slope map and MERRA-2's topographic slope map could be quite different. Please comment in the manuscript how this discrepancy impacts the calculation of wind directions and slopes in Fig. 1.

Line 187: It looks like this is the first appearance of the term super coarse dust. Please include a diameter range for super coarse dust.

Line 276: I think you can mention here that you eventually adopted the multiple linear regression (MLR) model.

Lines 305-307: For Fig. 6, can you color code the three panels for three wind directions by the value of wind speed? I am curious how wind speed varies with those vertically aligned scattered points.

Line 314: For Fig. 6, I am also interested in looking at a plot for soil moisture (can be in the supp).

Lines 339-340: Instead of seeing the relationship between coarse dust fraction and the 16 soil type/class, readers might be more interested to see the relationship between coarse dust fraction and clay fraction.

Line 360: For table 2, I am curious about the statistics of the five other predictor variables. Please include them in Table 2 or in the supp.

---

## Author Comment (AC1)

**Responses to Review Comments 1 by Anonymous Referee #2**

This article analyzes the association between the particle size distribution of windblown dust and topographical wind conditions over the Sahara, using linear regression models with inputs from the coarse fraction of dust from MONARCH dust reanalysis dataset, the wind conditions from MERRA2 meteorological reanalysis fields, and surface elevation data. Positive correlations between particle size and wind speed and uphill slope wind direction are found in this study. The scope of the manuscript is very important. However, I am concerned about the soundness of the method and thus the associated interpretation and conclusions.

Thank you for recognizing the significance of our study and providing valuable comments. We have addressed the concerns, particularly those regarding the relative coarse resolution of the reanalysis data, the representation of coarse fraction in MONARCH dust concentration, and the robustness of our statistical methodology. Please see our specific responses below.

**General Comments**

1. The linear regression model used in the study to state the relationship could be misleading and lack of strong evidence. The dust emissions are inherently nonlinear and vary with the cube (subjected to land surface properties) of surface friction velocity. The initiation of dust emissions is also subjected to the threshold friction velocity. The application of linear regression model and simple treatment of adding interaction terms between independent variables impose violation against the nonlinear processes in dust emissions, transport and deposition.

Dust emission models indeed reflected that dust emission flux exhibits a nonlinear relationship with the friction velocity and its threshold. However, the understanding of dust size distribution of dust remains comparatively limited. While models for dust size distributions exist (Kok, 2011) how the parameters may vary under changing meteorological conditions is understudied. Notably, there is ongoing debate about whether the influence of friction velocity on dust particle size is positive or negative (Shao, 2011; Kok, 2011; Malinovskaya et al., 2021). As such, the goal of this study is to explore the potential relationship between dust PSD and wind conditions by analyzing a larger amount of data, with the initial objective of determining the direction of this influence. Linear models are simple to understand and implement, yet they are robust and capable of this task. Moreover, they are highly interpretable that coefficients directly show the relationship between predictors and the outcome. In addition, we attempted alternative traditional non-linear regression models but the coefficients of determination ($R^2$) indicate that the alternative traditional regressions only have marginally improved performance due to the intrinsic heteroscedasticity of the data (Fig. 6). No apparent non-linear trends observed from the explanatory analysis further led us to proceed with the linear model. The linear models are not intended for suggesting linear relationships, but rather are used as a tool to analyze the sign and magnitude of the influence. The clarification was added to the main text: "These linear models are not intended to imply strictly linear relationships between dust PSD and wind conditions, but rather to provide initial guidance on the overall expected directionality of these relationships." In addition to the alternative traditional nonlinear models, we further constructed the machine learning models, whose prediction accuracy does not

outperform the linear models. We have added those results to increase the credibility of shared patterns observed in both linear and machine learning models.

2.  The poor explainability from the linear regression model without interaction terms ($R^2$ of 0.224) and with interaction terms ($R^2$ of 0.239) between independent variables questions the soundness of the results and interpretability. Thus, the interpretation from the manuscript is not based on strong evidence and at a worse case potentially causes misleading conclusions. The $R^2$ measures how much of the data variability is explained by the model and the low values were due to the heteroscedasticity (i.e., varying spread of the data across values) which cannot be easily captured by a simple model. The linear model does not perfectly fit all data points but captures the average trend in the data. Linear regression models can provide a clear understanding of the overall relationships while acknowledging some variability in individual points. Despite the variations for individual data points, our model effectively predicts the overall trend. The response vs. fitted value plots (Fig. S3) show that the predicted values closely align with observed values, clustering around the one-to-one red line. Additionally, we examined the prediction intervals for each data point and compared them with observed values. In both models, over 94% of these intervals contain the observed coarse fraction values, demonstrating strong predictive coverage. The main text was updated accordingly: "Although individual data points present deviations, our models effectively predict the overall trend, as suggested by the response vs. fitted value plots (Fig. S4), where the predicted values align closely with observed values and cluster around the one-to-one red line. Adjusted GVIF values for the model without regression were consistently below 2 with most values close to 1 (Table S2), indicating that multicollinearity among continuous or categorical predictors does not significantly affect the regression model."
    In addition to  alternative linear models with transformations, weighted linear regression, and beta regression, we further applied machine learning (ML) models.  Specifically, we constructed Random Forest and Extreme Gradient Boosting (XGBoost) models which are common choices for regression with optimized hyperparameters. Since no missing values are allowed in the model, we separated the slope and the wind direction type. The Random Forest and XGBoost models attain coefficient of determination ($R^2$) with 10-fold cross validation (CV) of 0.259 and 0.273, respectively, which present marginal improvements compared to linear models. Their prediction interval accuracy with 10-fold CV are 92.5% and 54.4%, which are lower than the 94% of linear models. We therefore adapted the linear models. Interpretations of the ML models were made possible by SHapley Additive exPlanations (SHAP); however, SHAP can only provide the direction of feature effects, not their magnitudes. The resulting summary plots from the SHAP analyses are shown in Fig. S5. The Random Forest model suggests that both higher wind speed and steeper slope are associated with higher fraction of coarse dust – these findings are consistent with the linear model without interactions; in terms of relative wind direction, uphill increases the dust size whereas tangential and downhill are correlated to higher fine dust fraction. The descriptions on methods and results of ML models have been added to the manuscript in Section 2.1 and Section 3.3. In a nutshell, linear models have competitive performance for our dataset compared with ML models.

3. The coarse resolution of input data and the validity of capturing fine-scale terrain induced wind fields and dust emissions are not strongly evaluated. The coarse resolution of MERRA2 at $0.5° \times 0.625°$ cannot resolve localized wind fields over regions with steep slopes, and the usage of 2-m winds from MERRA2 cannot represent actual localized 2-m winds due to elevation averaging. The manuscript is heavily based on the MONARCH dust reanalysis dataset. Although it has satellite assimilation embedded and shows generally agreement against regional mean measurements, it is still questionable to resolve the fine-scale dust emissions or concentrations. Considering the target of this manuscript over locations with prominent surface elevation changes, fine-scale variability is especially important to gain insights. Thus, the coarse resolution of independent variables, and questionable fine-scale validity of dependent dust concentrations from MONARCH can impose severe reliability of the interpretation and conclusions from this manuscript.

   Although the resolution of MERRA-2 reanalysis limits the ability to resolve fine-scale gusts or localized orographic wind channeling, the upsampled wind fields effectively represent the spatially averaged wind conditions that govern dust emissions over large regions. The primary signal of interest of this study is the broad-scale modulation of dust emissions induced by terrain, rather than the microscale fluctuations. In this context, the MERRA-2 wind fields are interpreted as representative of the effective wind forcing over each grid cell. The robust physical parameterizations on the MONARCH model and satellite-derived dust optical depth assimilation, complimented by validation against independent observations for fresh dust emission demonstrate the capability of the reanalysis in capturing dominant regional patterns. While we acknowledge the inherent limitations on resolution of the reanalysis datasets, the integration of data assimilation, upsampling techniques, and a focus on regional-scale patterns ensures that our conclusions regarding the impact of topography on dust variability remain interpretable. We added clarifications on this point to the main text: "While we acknowledge the inherent resolution limitations of reanalysis datasets, the focus of this study is on the broader-scale modulation of dust emission by wind conditions, and data assimilation combined with upsampling techiniques ensure that our conclusion remain interpretable in this context."

4. As said in the paper, MONARCH assimilates coarse dust optical depth (DOD) from satellite with fixed first-guess particle size distribution of emitted dust. How would the uncertainties for the assimilation of coarse DOD propagate into the particle size distribution of dust?

   Assimilating the coarse-mode DOD provides a strong constraint on the total coarse dust mass in the model and was used to adjust the dust concentration in the five coarse bins (1.2-20 μm). Although it is unclear how the increments were distributed among these bins, if the prior size partitioning is reasonably accurate, this correction will lead to a better representation of the coarse dust distribution relative to a purely free-running model. Nevertheless, there remain uncertainties in how those increments propagate into (or remain uncorrected in) the finer bins. Overall, while the coarse-DOD assimilation demonstrably enhances the particle size distribution beyond the first-guess, any significant biases in the prior size partition may still persist in the reanalysis.

5. The article uses coarse fraction of dust concentrations as a surrogate for the particle size distribution. This is an important piece of information. I recommend clarifying it in the abstract.

   The abstract has been revised to explicitly state that the coarse dust fraction was used as a surrogate for the dust size distribution: "...the fraction of coarse dust concentrations was calculated as a surrogate for size distribution."

**Specific Comments**

1. Line 101-103: what is the performance of first-guess particle size distribution of emitted dust compared to Fennec?

   We have included the first-guess size distribution in Fig. 3 for comparison. Additionally, the dust particles density used for volume-to-mass conversion has been updated to align with the MONARCH model (Klose et al., 2021). Overall, the dust concentration reanalysis shows improved agreement with observations compared to the first-guess estimates.

2. Line 109-111: How would the interpretation be sensitive to the definition of coarse fraction used here? For example, how would the results change when using coarse fraction as the mass of largest dust bin over the total mass of dust?

   To assess the sensitivity of our findings to the definition of coarse fraction, we compared the results using different coarse fraction metrics. In addition to the definition used in the study (mass fraction of the top two bins (6-20 μm, namely "cf2"), we also computed the coarse fraction for only the top bin (12-20 μm, namely "cf1") and for the top three bins (3.6-20 μm, namely "cf3"). Resulting coefficients of linear models without interactions constructed based on each definition are presented in Table S4. The results for cf1 shows distinct patterns, whereas results for cf2 and cf3 are largely consistent. Given that dust in the 12-20 μm bin falls into the "super coarse" dust category (Meng et al., 2022), this suggests that the super coarse dust responds differently to the varying wind conditions compared to other coarse dust. Since "super coarse" dust alone is not an adequate proxy for "coarse" dust in the context of regional-scale emission and transport, we conclude that cf2 or cf3 provide more suitable representations of the coarse fraction.

   Relevant statements have been added to the methods and results sections in the main text: "Additionally, we tested alternative definitions of coarse fraction (namely "cf1" and "cf3", where the coarsest one or three bins are assigned as coarse dust) in the subsequent statistical analysis and results suggest that cf2 is a representative surrogate for dust particle size. More details on the comparison are included in Table S4 and Section 3.3." and "As noted in Section 2.1, we also tested two alternative definitions of coarse fraction (cf1 and cf3) and compared the estimated coefficients with their statistical significance in Table S4. The coefficient estimates based on cf2 and cf3 are largely consistent, whereas those based on cf1 show some distinct patterns. Given that dust in the top bin (12-20 μm) falls into the "super coarse" dust category (> 10 μm; Meng et al., 2022), these results suggest that super coarse dust responds differently to varying wind conditions compared to other coarse dust. Therefore, the cf2 definition serves as a robust representation of coarse dust particles."

3. Line 186-189: From Figure 3, it looks like the size distribution for size bins of 0.6-12 $\mu$m is overestimated by MONARCH compared to Fennec. How would that affect the analyses for particle size distribution of dust?
The overestimations of dust concentrations in the 0.6-1.2 µm size range by MONARCH lead to an underestimation of the fraction of dust in the coarsest bin. However, this effect is mitigated by defining the coarse fraction based on a broader range of size bins. Sensitivity analysis using different definitions of the coarse fraction confirms the robustness of the chosen definition (see also Specific Comment 2).

4. For Figure 6, the points overlap with each other too much, making it hard to see clearly. Could it help to show the results with the number of points color coded?
Figure 6 has been improved by color-coding the scatter points based on the number of overlapping points in the upper left panel for wind speed. The updated plot displays condensed data points around the trendline, particularly at lower wind speeds.

***Responses to Review Comments 2 from Anonymous Referee #1***

General comments:

The manuscript titled "*Impact of Topographic Wind Conditions on Dust Particle Size Distribution: Insights from a Regional Dust Reanalysis Dataset*" by Huang et al. is a manuscript that attempts to investigate what factors change the emitted dust particle size distribution (PSD). They did this analysis by employing the MONARCH chemical transport model with the assimilated dust optical depth (DOD). The authors claimed that by assimilating MONARCH dust using observed DOD, MONARCH has the ability to produce adequate emitted dust PSD. The authors used the MONARCH surface dust concentrations to construct atmospheric dust concentration PSD, claiming this represent the emitted dust PSD. The authors tried to truncate aged transport dust and retain fresh dust by selecting only strongly and freshly emitting dust events using several criteria. Then, the authors performed a multiple linear regression (MLR) analysis to quantify the sensitivity of coarse dust fraction (coarse dust concentration divided by total dust concentration) to various meteorological (wind speed, wind direction) and land-surface variables (e.g., topography, soil texture, soil moisture). The authors found out a predictor model that shows that coarse dust fraction increases with wind speed and with hillslope. They also showed that some times of the day or some seasons favor the production of coarse dust.

This is an interesting and impactful study showing that the observed coarse dust might not be decreasing with or invariant with wind speed as some of the previous theories depicted (e.g., Shao, 2001; Kok, 2011). Therefore, the contribution of this paper is significant. The paper is also nicely written. However, since the paper is based on model/reanalysis and not observations, the authors need to show enough evidence to convince us that MONARCH produces correct dust PSDs. I have some major and minor comments below concerning some parts of the arguments and analysis, and I suggest a major revision.

Thank you for acknowledging the contribution of this paper and providing insightful feedback. We highly appreciate your comments and tried to address them below.

Major Comments:

1a. What I concern the most is that the findings from this paper are highly contingent upon how successfully MONARCH assimilates the fresh dust concentration PSD. All major findings of this paper are based not on observations but on the MONARCH reanalysis data. As the authors said, the correlation between coarse dust 'concentration' fraction and wind speed can be due to other reasons (e.g., turbulence can hold coarse dust longer in the atmosphere). Even though the authors tried to focus on fresh dust events to minimize the transport effect, the correlation could still be just because there are just more fresh dust particles that contains coarse dust, as the authors referred to Cristina's argument several times (Gonzales-Florez et al., 2023). This means that the correlation the authors found will only be valid for atmospheric dust concentration PSD, but not for dust emission PSD.

Clarifying the definition of "dust emissions" is indeed critical. To improve clarity, we updated the abstract and the summary of study in the Introduction to explicitly differentiate the concept of dust emission PSD from the dust concentration PSD: "A 10-year dust reanalysis underwent selective screening to identify events with fresh emissions and the fraction of coarse dust concentrations was calculated as a surrogate for size distribution" and "...constructed models to investigate the

correlations between PSD of surface dust concentrations in fresh emissions and wind conditions over slopes…”.

1b. Following the previous point, it is great that MONARCH shows this correlation between coarse dust concentration fraction and wind speed, but this is not only the evidence in concentration (not in emission) but also model evidence (not from in-situ measurements). So, the current evidence to me is not strong enough to reject Shao's saltation bombardment theory (Shao, 2001) or Kok's brittle fragmentation theory (Kok, 2011), which focused on the dust 'emission' PSD. But, at least, if MONARCH' dust PSD is correct, then other chemical transport models using Shao or Kok's theories should still replicate this correlation between coarse dust concentration and wind speed. We compared the observed trend with Shao's and Kok's theories in order to understand the underlying mechanisms behind the observed trends, and we agree that our results based on regional dust emissions (with a portion of transported dust) do not necessarily reject these theories which are validated against measurements in wind tunnels or over local sites. The MONARCH ensemble simulations adapted dust emission schemes developed by Marticorena and Bergametti (1995), Ginoux et al. (2001) with modifications, and Kok et al. (2014) (Klose et al., 2021), which should be comparable to simulations from other chemical transport models. We rephrased the discussion to emphasize that  the observed trend is contradictory to the implications of the theory, instead of the theory itself, as well as to highlight that “… since the fresh dust emissions at regional scale inevitably include transported dust, yet another potential explanation is unrelated to the emission, but to the transport process…”.

2. Line 103: Why do you think that MONARCH, by only assimilating coarse-mode DOD (and not finemode DOD), is enough to fully constrain the whole dust PSD? (I surely know we can hardly isolate fine dust from other fine-mode aerosols in MODIS). The PSD curve has two ends, and MONARCH only assimilates coarse-mode DOD, which constrains the right/high end. There is no information to constrain the left/low end, and MONARCH could be underestimating fine dust by not assimilating fine-mode dust. The correlation between coarse dust and wind speed might be a spurious artifact from MONARCH.
We appreciate this important question. In principle, omitting an explicit assimilation of fine-mode DOD does introduce some uncertainty regarding the fine end of the dust size distribution. Nonetheless, the MONARCH data assimilation framework and its physical parameterizations exert constraints across all dust size bins, including fine-mode dust, through the following procedures: In MONARCH, the dust state vector being updated through the 4D-LETKF assimilation includes the total coarse dust mixing ratio, which is distributed across five size bins from 1.2 to 20 μm (Di Tomaso et al., 2022). Although the assimilated observations represent exclusively coarse-mode DOD, the model's internal representation ties adjustments in the coarse bins to the overall dust mass and emission fluxes. Therefore, any increment that corrects coarse-mode dust fields effectively modifies total dust emissions, transport, and deposition in ways that also affect smaller particles. Further, the fine dust bins (0.2–1.2 μm) are not treated as independent of the coarse bins. After the assimilation step, increments in coarse dust are consistently “scaled down” to the fine bins (proportional to their relative mass in the model) so that the overall partitioning remains physically plausible. Although this approach does not fully replace assimilating fine-mode retrievals, it prevents fine-mode dust from being unbounded or disconnected from the coarse-mode reanalysis. Thus, in practice, all eight dust bins (0.2–20 μm) are updated. Finally, the ensemble-based 4D-LETKF simultaneously optimizes the coarse dust field at each time window using flow-dependent background error covariances. Hence, if fine bins become inconsistent with

the observed coarse dust or the model's internal aerosol processes, the ensemble spread (which includes perturbations in emission fluxes and size distribution) allows the assimilation to reduce the gap to reach physically consistent states. This ensemble-driven error covariance is a key reason why even partial observational constraints can significantly improve unobserved parts of the state vector. Ultimately, we acknowledge that residual uncertainties remain in the exact fine-to-coarse partition and additional constraints can be beneficial for capturing sub-micrometer dust more precisely, but validation against AERONET observations suggest that fine dust is adequately captured. To clarify this, we highlighted and re-emphasized this point in the manuscript: "...although MONARCH reanalysis does not directly assimilate fine-mode DOD, corrections in the coarse bins propagated to the entire PSD through the assimilation state vector and physical parameterizations, aligning the PSD more closely with dust-specific observations. Validation against AERONET data indicates that fine dust is still captured satisfactorily (Di Tomaso et al., 2022; Mytilinaios et al., 2022), supporting the reliability of the dataset to investigate dust PSD".

3. Lines 106-107: Following the first point, dust surface concentration PSD is not dust emission PSD, although they tend to correlate more in fresh dust events. It inevitably includes also transported dust, and so using dust concentrations PSD already underestimates coarse dust particles. Does this partially weaken the slope and $R^2$ you found?

It is indeed possible for the deposition of dust particles to affect the variation of coarse fraction and thus the $R^2$. The weak correlation may indicate the presence of factors - in addition to deposition and transport - that are not accounted for in the model, such as variability within the same soil texture type or differences in the nature of emission events occurring at similar times of day. We updated the main text to comment on this point: "The somewhat weak correlations may be related to confounding factors that are not included in the model, such as deposition and variability within the same categories of soil texture."

4. Lines 176-180: I am curious about Claire Ryder's FENNEC data. Since they have several FENNEC flights, did the authors see a correlation between the coarse dust fraction from her fresh dust PSD and wind speed? It would be better to look at observed correlations too.

During the Fennec campaign, dust was sampled through wind-mounted instruments on aircrafts at altitudes of approximately 1-6 km. Therefore, the dust PSD measurements were subjected to transport and deposition, making them unsuitable for analyzing the relationship between wind conditions and fresh dust emissions. For example, among the three flights selected for evaluation in our study, the b600, which was conducted in early morning after notable dust uplift by low-level jets, detected the coarsest dust during the entire campaign of up to 300 μm, whereas dust collected from the b601 and b602 flights, conducted in the afternoon and the following morning, contained less pronounced coarse dust due to vertical mixing and deposition (Ryder et al., 2015).

5a. Line 189: Since the whole paper's findings is contingent upon the accuracy of MONARCH's assimilated dust concentration PSD, I am not sure if one plot (Fig. 3) of MONARCH vs FENNEC is enough for model evaluation. Fig. 3 only shows that MONARCH's simulating adequate coarse dust, but it does not show that the posterior (assimilated PSD) is doing better than the prior (first guess PSD), and it is hard to judge from Fig. S1. Could you include the first guess in Fig. 3 in a different color and describe how the assimilated PSD is better than the first guess?

We have revised Figure 3 (copied here) to include the first-guess size distribution for a clearer comparison between the prior and the reanalysis. While a comprehensive evaluation of MONARCH reanalysis is beyond the scope of this study, we particularly examined one case to show that the predicted dust concentration falls within the reasonable range compared to observations of freshly

emitted dust. Further details on the performance of MONARCH reanalysis can be found in Di Tomaso et al. (2022) and Mytilinaios et al. (2022). The main text has been updated accordingly to address this point: "The MONARCH dust reanalysis dataset was previously evaluated against observations from the Aerosol Robotic Network (AERONET) retrievals (Di Tomaso et al., 2021; Mytillinaios et al., 2023).... Here, we present an additional case study to particularly evaluate the performance of dust reanalysis on capturing fresh dust emissions."

[Figure]

**Figure 3**. The black line shows the average volumetric concentration ($\mu m^3$ $cm^{-3}$) of dust sampled during three Fennec flights (6 flight segments) and the grey shaded area denotes the range of values. The blue bars show the volumetric concentration ($\mu m^3$ $cm^{-3}$) of dust in corresponding grids from the MONARCH reanalysis calculated from the weighted average mass concentration, with a particle density of 2,500 kg $m^{-3}$ for the finer four bins and 2,650 kg $m^{-3}$ for the coarser four bins. The error bars denote the range of values.

5b. Moreover, this evaluation in Fig. 3 does not show that the correlation between coarse dust fraction and wind will hold true in observations.
Fig.3 was intended to demonstrate that MONARCH reanalysis reasonably captures the PSD for fresh emissions, but was not meant to serve as a comprehensive evaluation. We are unable to conduct robust statistical analysis based on merely measurements due to its limited availability, and we utilize the MONARCH reanalysis to investigate the relationship between dust size and wind conditions with full awareness of its uncertainties.

6. Lines 115-117: I am concerned that this paper would yield very different regression results if the authors used MONARCH's predictor variables rather than MERRA-2's fields. Please describe how MONARCH has its own driving meteorology. Did they do any meteorological assimilation or was it a complete free run? Please also briefly describe how we should expect MONARCH's met fields to be different from MERRA-2's met fields, and the implications on your regression analysis (e.g., will the sensitivities of coarse dust to wind directions largely enhance using MONARCH's meteorology?)
The MONARCH dataset estimated dust emissions using an ensemble where the meteorological inputs were perturbed. Specifically, the wind fields were re-initialized daily using reanalysis from either MERRA-2 dataset (for 6 out of the 12 ensemble members) or ERA-Interim dataset (for the other 6 ensemble runs; see Table S1 in Di Tomaso et al., 2021). Previous evaluations suggest that the meridional and zonal winds in these reanalysis are strongly constrained by observations

(Fujiwara et al., 2024; Rienecker et al., 2008). The observational data assimilated by ERA-Interim were all included and made up the majority of the assimilation data used for MERRA-2 – including wind observations from the ERS-1, ERS-2, QuikSCAT datasets (Dee et al., 2011; Gelaro et al., 2017). Furthermore, performance of both reanalysis were pretty similar over specific sites (Santos et al., 2019). In general, considering that the MONARCH's dust emission partially rely on meteorological inputs from MERRA-2 and the comparable results between MERRA-2 and ERA-Interim, it is reasonable to use MERRA-2 wind variables to represent the wind conditions in the MONARCH reanalysis.

The main text was updated accordingly: "MONARCH ensemble simulations applied meteorological inputs from two reanalysis datasets (i.e., MERRA-2 and ERA-Interim). Given that wind from both reanalyses are highly constrained by observations, and there is a substantial overlap in the assimilated data used by the two (Fujiwara et al., 2024; Rienecker et al, 2008; Dee et al., 2011; Gelaro et al., 2017) , it is reasonable to use MERRA-2 wind vectors to inform the wind conditions of MONARCH dust reanalysis".

7a. Line 255-257: Since year, season, and time of day are correlated with winds and soil moisture, adding them in your regression analysis likely weakens the $R^2$ of winds and soil moisture on dust PSD. Please comment on how this collinearity between predictor variables impacts the regression results.

Collinearity or multicollinearity among predictor variables can inflate standard errors and reduce the statistical significance of regression coefficients. To assess this, we calculated the Generalized Variance Inflation Factors (GVIFs) for all predictors in our model without interactions (Eq. (3)) using the VIF function in R software, and then adjusted the GVIF by the degree of freedom (Df) of categorical variables (adjusted GVIFs are expressed as $GVIF^{\frac{1}{2 \cdot Df}}$). An adjusted GVIF of 1 (the smallest) indicates no collinearity and typically values smaller than 5 suggest low and acceptable collinearity. Our results show that all adjusted GVIF values were below 2 with most values close to 1 (Table S2, copied below), indicating that multicollinearity is not a concern in this regression model.

We updated the main text accordingly: "Collinearity or multicollinearity among predictors can inflate standard errors and reduce the statistical significance of regression coefficients. To assess this, we calculated the Generalized Variance Inflation Factors (GVIFs) for all predictors in the linear models using the VIF function in R (R. Core Team, 2023). For categorical variables, the GVIFs were adjusted by the degrees of freedom (Df), expressed as $GVIF^{1/(2 \cdot Df)}$). An adjusted GVIF of 1 (the smallest value) indicates no collinearity, while values below 5 generally suggest low and acceptable collinearity.... Adjusted GVIF values for the model without regression were consistently below 2 with most values close to 1 (Table S2), indicating that multicollinearity among continuous or categorical predictors does not significantly affect the regression model."

**Table S2.** Generalized variance inflation factors (GVIFs) for all predictors in the model without interactions (Eq. (3)). The GIF adjusted for the degree of freedom (Df) values ($GVIF^{1/(2 \cdot Df)}$) being 1 (the minimum) indicates no collinearity and values smaller than 2 typically suggest low and acceptable collinearity.

| Variables | GVIF | Df | $GVIF^{1/(2 \cdot Df)}$ |
|---|---|---|---|
| wind speed | 1.139 | 1 | 1.067 |
| slope with uphill winds | 1.998 | 1 | 1.413 |
| slope with tangential winds | 2.654 | 1 | 1.629 |
| slope with downhill winds | 2.187 | 1 | 1.479 |
| time of day | 1.248 | 2 | 1.057 |
| season | 1.453 | 3 | 1.064 |
| year | 1.004 | 1 | 1.002 |
| soil moisture | 1.313 | 1 | 1.146 |
| soil texture | 1.651 | 8 | 1.032 |

7b. Following the previous comment, wind speed and soil moisture are highly correlated. Including soil moisture likely highly changes the value of $\beta_1$.

We examined whether soil moisture in the regression model significantly alters the estimated coefficient for wind speed ($\beta_1$) using the Pearson correlation coefficient between wind speed and soil moisture and the adjusted GVIF values. The correlation coefficient was -0.161 and the adjusted GVIFs for both variables were low, suggesting that the potential inflation of $\beta_1$'s standard error is minimal and acceptable.

8. Lines 313: I agree with another reviewer that the $R^2$ of the regression analysis is rather low. From Fig. 6, it looks like to me that instead of a spatial regression analysis, it might be better construct regression models for each grid/small region rather than putting them together into a single plot.

We agree that the spatial variations in dust size should be considered when constructing the model, but defining meaningful sub-domains presents challenges due to the complex and continuous nature of soil properties. Therefore, we decided to incorporate soil texture as an independent variable into our regression models for the entire study domain considering that the differences among geographic locations are mostly related to soil properties. To test the sensitivity of the models to spatial variability, we performed 10-fold cross-validations, in which the dataset was randomly split into 10 even subsets. The model was trained on 9 subsets and tested on the remaining subset in each iteration, with 10 iterations in total till all subsets were used for testing. This approach yields coverage rates of 94%, suggesting that 94% of the coarse fraction values from MONARCH reanalysis in the test subset are covered by the prediction intervals of the model . The low $R^2$ suggests high variability in the data due to the heteroscedasticity as shown in Fig. 6, and separate models constructed for sub-domains may reduce the $R^2$ due to smaller sample sizes and

variability. However, the high coverage rates from cross-validation suggest that the observed trends remain generally invariant across different sub-domains.

9. Lines 364-365: This statement does not sound correct to me. A multiple linear regression (MLR) analysis does not give the slope/sensitivity of a predictor variable while holding other predictor variables constant, i.e., it is not a partial differential $\frac{\partial(coarse\ dust)}{\partial(slope)}$. The magnitude of the slope from MLR is heavily impacted by the multicollinearity among different predictor variables. It is all right to say that coarse dust increases with slope regardless of wind direction, but other factors are not held constant In the MLR when this slope is estimated. Please clarify this sentence.

While MLR models estimate the effect of each predictor while accounting for others, it does not compute a partial derivative $\frac{\partial(coarse\ dust)}{\partial(slope)}$ in the calculus sense. The regression model is stochastic rather than deterministic, so its interpretation does not rely on partial derivatives. Instead, the regression coefficient represents the expected change in the dependent variable given a unit change in the predictor, assuming all other variables remain at constant values. The magnitude of regression slope can be influenced by multicollinearity among predictors. However, as noted in our response to Comment 7a, the GVIF analyses indicate no evidence of multicollinearity among the independent variables, suggesting that interpreting the trends in coarse fraction from the regression coefficients is reasonable. To avoid ambiguity, we updated the text: "The coefficient for wind speed from the linear model is statistically significant and positive (Table 2), suggesting that controlling the effects of other factors, the coarse fraction of dust emissions increases with the wind speed." Note that the estimated slopes (coefficients) are subject to interactions among variables even though the regression model accounts for effects of other predictors.

10. Lines 372-376: The model shows that regardless of wind direction (uphill, tangential, downhill), coarse dust fraction increases with slope. Normally we will expect if coarse dust increases with more uphill slope, then the opposite should be true (fine dust increases with more downhill slope). The three explanations to why increasingly uphill, tangential, and downhill slopes can all generate more coarse dust fraction sound a little ad hoc. Under what circumstance should the fine particle fraction increase?

Due to the complex nature of wind flows over varying terrain, it is difficult to predict the net effects of slope on dust particle size from the physical basis, and we do not necessarily expect the effects of uphill and downhill winds to be opposite. Instead, both our linear models and the Random Forest model suggest a consistent positive correlation between slope and coarse fraction, regardless of wind direction (Table 2 and Fig. S5). However, the impact is stronger for uphill winds than tangential or downhill winds, as the Random Forest model suggests negative impacts of these directions. These updates have been reflected in the final paragraph of Section 3.3.

Other comments:

Lines 72-74: Does this finding come from Malinovskaya 2021? Please cite if so. Does this conflict this paper's findings?

Yes, the finding comes from Malinovskaya et al. (2021) and we have added the reference. Their conclusion about the influence of wind direction on dust size is opposite to our findings. However, this does not necessarily reject their theory, and may instead indicate that other factors (such as near-source transport) overshadow the impacts of microphysics of dust emission at a regional

scale. This point has been addressed in the main text: "Our results suggest that at the regional scale, the effect of near-source transport of emitted dust at scales of hundreds to thousands of meters dominates over the impact of microphysics of dust emission related to secondary dune structure at scales of centimeter to meters."

Lines 97-99: A reference is needed on how coarse-mode DOD was obtained from Aqua/MODIS, so people do not need to go to MONARCH's paper to look for it. Is it Paul Ginoux's method?

We have added the references for the derivation of the coarse-mode DOD: "The assimilation data of coarse-mode dust optical depth (DODcoarse) were derived from the Moderate Resolution Imaging Spectroradiometer (MODIS)-Aqua Deep Blue level 2 aerosol products (Collection 6), including aerosol optical depth (AOD), the Ångström exponent, and the single scattering albedo at different wavelengths (Ginoux et al., 2012; Pu and Ginoux, 2016)." The method follows the derivation of DOD in Ginoux et al. (2012), with adaptations for MODIS Collection 6 data as described in Pu and Ginoux (2016).

Lines 100-102: What prior (first guess) dust PSD does MONARCH assume? This requires a more detailed description here. How much is the posterior dust PSD (after assimilation) dependent on the assumed prior dust PSD? This question also needs some elaboration.

MONARCH's first-guess dust PSD is based on a sectional representation of dust across eight bins spanning from 0.2 to 20 μm in diameter. The assumed emission-size distribution (PSD) follows the brittle-fragmentation theory of Kok (2011) with perturbations across 12 ensemble members (Table S1 in Di Tomaso et al, 2022). Because only coarse-mode dust optical depth is assimilated, the observational constraint applies to the aggregate dust mass in bins covering approximately 1.2–20 μm. The partition of increments among dust bins are implicit, and the total mass of finer modes (0.2-1.2 μm) is adjusted proportionally based on the coarse-to-fine ratio in the first-guess. Consequently, if the prior PSD is biased—for instance, by allocating too much mass in the largest bin or not enough in a mid-sized bin—that bias may persist to some extent after assimilation. Despite the limitation, assimilating DODcoarse effectively reduces large-scale biases in coarse dust load compared to a free-running model and can improve the dust size distribution, provided that the prior size partitioning is not severely erroneous.  The procedures of adjusting dust PSD through assimilation and associated uncertainties are also elaborated in our response to Major Comment 2. To clarify this point in the manuscript, we have added the following statements: "MONARCH's first-guess dust size distribution follows the brittle-fragmentation theory of Kok (2011) with perturbations across 12 ensemble members", and "Consequently, if the prior PSD is biased—for instance by placing too much mass in the largest bin or not enough in a medium bin—that bias may persist to some extent after assimilation."

Line 103: Did MONARCH "nudge" the size-resolved concentrations or used a Kalman filter? They are different methods.

MONARCH reanalysis of the dust mixing ratio in the five coarse bins was generated using the four-dimensional Local Ensemble Transform Kalman Filter (4D-LETKF).  The term "nudge" was used loosely in the sense of "pushing toward". We agree that it's confusing and to avoid ambiguity, we have rephrased the description for the assimilation methods: "By applying a local ensemble transform Kalman filter with four-dimensional extension (4D-LETKF) at each 24-hour assimilation window, reanalysis increments were added to the model ensemble simulations (first-guess) to match the DODcoarse observations."

Lines 129-131: A description of how MERRA-2's assumed topography can alter MERRA-2's winds is needed. MERRA-2 has a coarse grid resolution of 0.5°, so the winds can only see a regional topographic slope. Also, I think that SRTM GL3's topographic slope map and MERRA-2's topographic slope map could be quite different. Please comment in the manuscript how this discrepancy impacts the calculation of wind directions and slopes in Fig. 1.

The MERRA-2 wind reanalysis is generated through the Goddard Earth Observing System (GEOS) atmospheric model with data assimilation (Gelaro et al., 2017). Wind vectors at 2m height are interpolated from wind simulations in the lowest model layer using the Monin-Obukhov equation. MERRA-2 topography is derived from the USGS Global 30 arc-second elevation (GTOPO30) dataset, and sub-grid topographic slope and variation were statistically represented to account for gravity waves and turbulence-induced wind adjustments (Rienecker et al., 2008).

Since both GTOPO30 and SRTM are seamless elevation dataset available at resolution much finer than that of MERRA-2 grids (Gesch et al., 2001), their spatial averages are expected to be comparable. Moreover, MERRA-2 wind vectors are strongly-constrained by assimilated wind observations from various sources (Fujiwara et al., 2024; Rienecker et al, 2008), further reducing their sensitivity to underlying topographic inputs. Overall, discrepancies between SRTM and MERRA-2 topography should have a limited impact on the wind reanalysis. We added the clarifications to the manuscript:

"The SRTM elevation data are expected to be compatible with the MERRA-2 wind reanalysis because 1) both the SRTM and GTOPO30, which is used in the Goddard Earth Observing System (GEOS-5) model (MERRA-2's first-guess), have much finer than MERRA-2's grid, making their spatially averages comparable, and 2) MERRA-2 wind reanalysis are highly constrained by assimilated observations, reducing its dependency on topographic input."

Line 187: It looks like this is the first appearance of the term super coarse dust. Please include a diameter range for super coarse dust.

Super coarse dust is defined as dust particles with diameters above 10 µm (Meng et al., 2022). This definition has been added to the manuscript: "...MONARCH reanalysis is effective at capturing the coarse to super coarse modes (defined as dust with diameter greater than 10 µm; Meng et al., 2022)...".

Line 276: I think you can mention here that you eventually adopted the multiple linear regression (MLR) model.

We have updated the main text to explicitly mention the selection of multiple linear regression models at the end of Section 2.4 for Statistical Analysis: "Ultimately, the more complex models did not substantially outperform the multiple linear models. Given their high interpretability, ability to capture interactions between predictors, and competitive performance, we selected linear models for further analysis. SHAP results from the machine learning models were also included to compare key findings. Additional details on model performance and results are provided in Section 3.3."

Lines 305-307: For Fig. 6, can you color code the three panels for three wind directions by the value of wind speed? I am curious how wind speed varies with those vertically aligned scattered points.

The three panels for slopes in Fig. 6 have been updated, with scatter points now color-coded by wind speed associated with the dust events. The vertically aligned points generally display varying colors, indicating that dust events at the same location or locations with the same slope experienced different wind speeds.

Line 314: For Fig. 6, I am also interested in looking at a plot for soil moisture (can be in the supp).
The scatter plot of coarse fraction versus soil moisture, including a trend line, has been included in
Fig. S2 (upperleft panel). The trend line shows a negative relationship, suggesting that an increase
in soil moisture corresponds to a lower fraction of coarse dust. This finding aligns with the results
from the linear model without interactions (Table S3) but is counterintuitive. This discrepancy may
be due to interactions with other variables, as suggested by the model with interactions (Eq. 4),
where the coefficient for soil moisture is positive.

[Figure]

**Figure S2.** Scatter plots and linear trend lines of relationships between the coarse fraction of surface dust concentration
and soil moisture and slope under three different wind directions. The color-codes presents the number of overlapping
data points.

Lines 339-340: Instead of seeing the relationship between coarse dust fraction and the 16 soil
type/class, readers might be more interested to see the relationship between coarse dust fraction
and clay fraction.
Since our soil texture variable has a category for clay and also accounts for the spatial variations in
chemical composition, we decided to adhere to the categorical predictor of soil texture.

Line 360: For table 2, I am curious about the statistics of the five other predictor variables. Please
include them in Table 2 or in the supp.
Please refer to Table S3 for coefficients for all the independent variables.

**References**

[revised manuscript text omitted]